# DEEP BAYESIAN FILTER FOR BAYES-FAITHFUL DATA ASSIMILATION

## ABSTRACT

State estimation for nonlinear state space models (SSMs) is a challenging task. Existing assimilation methodologies predominantly assume Gaussian posteriors on physical space, where true posteriors become inevitably non-Gaussian. We propose Deep Bayesian Filtering (DBF) for data assimilation on nonlinear SSMs. DBF constructs new latent variables $h_t$ in addition to the original physical variables $z_t$ and assimilates observations $o_t$. By (i) constraining the state transition on the new latent space to be linear and (ii) learning a Gaussian inverse observation operator $r(h_t|o_t)$, posteriors remain Gaussian. Notably, the structured design of test distributions enables an analytical formula for the recursive computation, eliminating the accumulation of Monte Carlo sampling errors across time steps. DBF trains the Gaussian inverse observation operators $r(h_t|o_t)$ and other latent SSM parameters (e.g., dynamics matrix) by maximizing the evidence lower bound. Experiments demonstrate that DBF outperforms model-based approaches and latent assimilation methods in tasks where the true posterior distribution on physical space is significantly non-Gaussian.

## 1 INTRODUCTION

Data assimilation (DA) is a crucial technique across various scientific domains. Its primary objective is to estimate the trajectory and current state of a system by integrating an imperfect model with partially informative observations. Specifically, given a series of observations $T$ time steps $o_{1:T}$, the goal is to infer the posterior distribution of the system's physical variables $z_t$: $p(z_t|o_{1:t})$. DA has been widely applied in fields such as weather forecasting (Hunt et al., 2007; Lorenc, 2003; Andrychowicz et al., 2023), ocean research analysis (Ohishi et al., 2024), sea surface temperature prediction (Larsen et al., 2007), seismic wave analysis (Alfonzo & Oliver, 2020), multi-sensor fusion localization (Bach & Ghil, 2023), and visual object tracking (Awal et al., 2023).

A key challenge in DA arises from the non-Gaussian nature of the posterior distributions $p(z_t|o_{1:t})$, which results from the inherent nonlinearity in both the system dynamics and observation models. Despite this, many operational DA systems, such as those used in weather forecasting, rely on methods like the ensemble Kalman Filter (EnKF) (Evensen, 1994; Bishop et al., 2001) for sequential state filtering (i.e., $p(z_t|o_{1:t})$) and the four-dimensional variational method (4D-Var) for retrospective state analysis (i.e., $p(z_t|o_{1:T}), t < T$). These approaches assume Gaussianity in their test distributions $q(z_t|o_{1:t})$ or $q(z_t|o_{1:T})$, a simplification driven by computational constraints. While exact methods such as bootstrap Particle Filters (PF) or sequential Monte Carlo (SMC) (Chopin & Papaspiliopoulos, 2020; Daum & Huang, 2007; Hu & van Leeuwen, 2021) could compute the true posterior, their performance degrades significantly when the number of particles is insufficient (Beskos et al., 2014). This issue is exacerbated in high-dimensional systems, making SMC approaches impractical for many physical problems.

To address these limitations, we propose a novel variational inference approach called Deep Bayesian Filtering (DBF) for posterior estimation. Our strategy consists of two main components: (i) constraining the test distribution to remain Gaussian to ensure computational tractability, and, in cases where the original dynamics are nonlinear, (ii) leveraging a nonlinear mapping to enhance the expressive capability of the test distribution. The DBF methodology diverges into two paths depending on the nature of the system dynamics, whether linear Gaussian or nonlinear:

**Linear dynamics** When the system's dynamics $p(z_{t+1}|z_t)$ are linear, DBF assumes Gaussianity in the original space, similar to traditional methods. However, DBF introduces the concept of the inverse observation operator (IOO; see also Frerix et al. 2021) to construct Gaussian test distributions $q(z_t|o_{1:t})$. The IOO, along with any unknown system parameters, are trained to minimize the Kullback-Leibler divergence between the test distribution $q(z_t|o_{1:t})$ and the true posterior $p(z_t|o_{1:t})$. The IOO and the system parameters are trained without teacher signals $z_t$.

**Nonlinear dynamics** In the more common case of nonlinear dynamics, DBF operates in a latent space, assuming Gaussianity in the latent variables $h_t$. The original physical variables are recovered through a nonlinear mapping function $\phi$, implemented via neural networks (NNs). This nonlinear mapping allows for a more flexible representation of the test distribution $q(z_t|o_{1:t})$. The IOO and other parameters are trained in a supervised manner (i.e., $z_t$ is used during training).

For state space models (SSMs) with nonlinear dynamics, DBF functions as a variational autoencoder (VAE) that adheres to the Markov property. Posterior distributions of the latent variables $h_t$ are expressed in a Bayesian framework. This approach is closely related to dynamical VAEs (DVAEs, Girin et al. 2021 for a review), which use VAEs to model time-series data. However, DBF distinguishes itself by its posterior design. Unlike DVAEs, where Monte Carlo sampling is required for inference (see Sec. 2.6.1), DBF allows for the analytical computation of the prediction step, recursively computing posteriors through closed-form expressions.

When applied to problems with nonlinear or unknown dynamics, DBF can be interpreted as learning the Koopman operator (Koopman, 1931) using NNs. The discovery of such latent spaces and operators through machine learning has been extensively studied (Takeishi et al., 2017; Lusch et al., 2018; Azencot et al., 2020) and will be experimentally validated through the handling of nonlinear filtering tasks involving chaotic dynamics.

Key contributions of the proposed DBF methodology include:

- DBF is the first VAE-based model for time-series data that maintains a posterior structure faithful to the Markov property in SSMs.
- For linear dynamics, DBF extends the Kalman Filter (KF) with a more flexible observation update. A simple object-tracking experiment is presented in Sec. 3.1. Additional experiments in Sec. B demonstrates that DBF can infer unknown system parameters via training.
- For nonlinear dynamics, DBF constructs a new latent space for data assimilation, allowing for the analytical integration of time steps and preventing the accumulation of Monte Carlo sampling errors. This is accomplished through the application of Koopman operator theory, which ensures that the model's representational power is maintained, as long as the latent space is sufficiently high-dimensional (see Sec. 3.2 and 3.3).
- As a generative model, DBF estimates the uncertainty of the physical variables $z_t$, in contrast to 3D- and 4D-Var, which yield only point estimates (see Sec. 3.2 and Fig. 3).
- The linear constraint on dynamics stabilizes the training process, which is known to be unstable in standard recurrent NNs (see Sec. 3.3 and Fig. 6).

DBF has demonstrated superior performance over classical DA algorithms and latent assimilation methods in scenarios with highly non-Gaussian posteriors, particularly in the presence of strongly nonlinear observation operators or large observation noise.

## 2 METHOD

### 2.1 INFERENCE OF PHYSICAL VARIABLES IN A STATE-SPACE MODEL

A physical system is defined by variables $z_t$, with its evolution described by the dynamics model $p(z_{t+1}|z_t) = \mathcal{N}(z_{t+1}; f(z_t), Q)$, where $\mathcal{N}(x|\mu, \Sigma)$ denotes a Gaussian whose mean and covariance are $\mu$ and $\Sigma$. The nonlinear function $f$ is the dynamics operator and $Q$ is the system covariance. The Markov property holds, as $z_{t+1}$ depends only on $z_t$. An observation model $p(o_t|z_t) = \mathcal{N}(o_t; h(z_t), R)$ relates observations to physical variables via the observation operator $h$ and covariance $R$. The panel (a) of Fig. 1 shows the system's graphical model. The objective of sequential DA is to compute the posterior of $z_t$ given $o_{1:t}$.

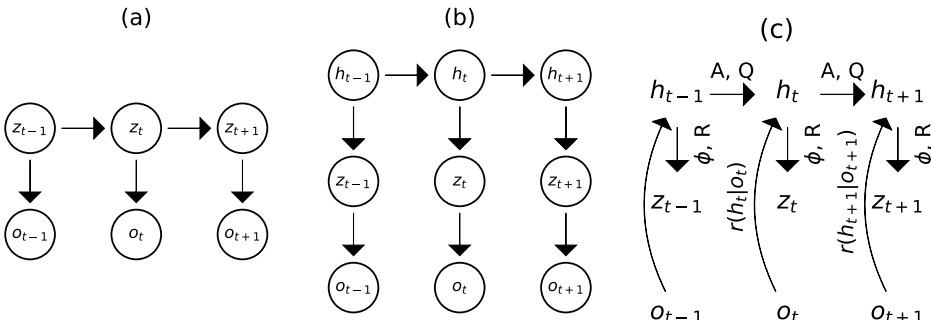

Figure 1: Panel (a) shows the graphical model for the simplest SSM. If the dynamics of the original SSM is linear, DBF assimilates on that space. Panel (b) shows the graphical model for the SSM assumed for SSM with nonlinear dynamics. Panel (c) shows the inference structure of our methodology for SSM with nonlinear dynamics.

## 2.2 KF FOR LINEAR DYNAMICS, LINEAR OBSERVATIONS

In the KF, the dynamics and observation models are both linear Gaussian. Given that the dynamics and observation operators $f, h$ are linear, we can represent them using matrices $A$ and $C$, respectively. All matrices ($A, C, Q$, and $R$) are constant. The filter distribution $p(z_t|o_{1:t})$ remains Gaussian, provided that the initial distribution $p(z_1)$ is Gaussian. We can recursively compute the posterior parameters (means $\mu_t$ and covariance matrices $\Sigma_t$) using the following equations:

$$\mu_t = \Sigma_t (A\Sigma_{t-1}A^T + Q)^{-1}A\mu_{t-1} + K_t(o_t - HA\mu_{t-1}), \tag{1}$$
$$\Sigma_t^{-1} = (A\Sigma_{t-1}A^T + Q)^{-1} + HR^{-1}H^T, \tag{2}$$

where $K_t = (A\Sigma_{t-1}A^T + Q)H^T(H(A\Sigma_{t-1}A^T + Q)H^T + R^{-1})^{-1}$ is the Kalman Gain.

## 2.3 DBF FOR LINEAR DYNAMICS, NONLINEAR OBSERVATIONS

In this scenario, Gaussianity of the test distribution is lost during the KF update step. We introduce an inverse observation operator (IOO) $r(z_t|o_t)$ (see also Frerix et al. 2021):

$$p(z_t|o_{1:t}) = \frac{p(o_t|z_t)p(z_t|o_{1:t-1})}{p(o_t|o_{1:t-1})} \propto \frac{r(z_t|o_t)}{\rho(z_t)}p(z_t|o_{1:t-1}), \tag{3}$$

where $r(z_t|o_t) = \frac{p(o_t|z_t)\rho(z_t)}{\int p(o_t|z_t)\rho(z_t)dz_t}$ and $\rho(z_t)$ is a prior virtually introduced for the IOO. By approximating both the IOO and the virtual prior as Gaussians, $r(z_t|o_t) = \mathcal{N}(f_\theta(o_t), G_\theta(o_t))$ and $\rho(z_t) = \mathcal{N}(m, V)$, respectively, the posterior $q(z_t|o_{1:t})$ can be analytically computed as a Gaussian, where the mean $\mu_t$ and covariance $\Sigma_t$ are given as:

$$\mu_t = \Sigma_t(A\Sigma_{t-1}A^T + Q)^{-1}A\mu_{t-1} + G_\theta(o_t)^{-1}f_\theta(o_t) - V^{-1}m, \tag{4}$$
$$\Sigma_t^{-1} = (A\Sigma_{t-1}A^T + Q)^{-1} + G_\theta(o_t)^{-1} - V^{-1}, \tag{5}$$

where $f_\theta(o_t)$ and $G_\theta(o_t)$ are NNs with parameters $\theta$, and $m$ and $V$ are constants set to $m = 0$ and $V = 10^8 I$. These values bias the NNs' outputs without affecting performance. The initial distribution $q(z_1)$ is taken to be a Gaussian with $\mu_1 = 0$ and $\Sigma_1 = 100I$.

The recursive formula for the exact posterior (Equation 3) requires no approximation. Thus, DBF computes the exact posterior when the true IOO $r_{\text{true}}(h_t|o_t)$ is Gaussian, i.e., the SSM is a Linear-Gaussian State Space (LGSS). In that case, the posterior update formula agrees with the KF (see Equations 1, 2 and 4, 5). The key difference is that nonlinear functions are applied to both the mean, $f_\theta(o_t)$, and the covariance, $G_\theta(o_t)$. In the KF, $f_\theta(o_t)$ is linear, and $G_\theta(o_t)$ is a constant matrix (see Equations 1 and 2). $G_\theta(o_t)$'s dependence on observations allows flexible adjustment of the new observation's impact on state estimation. The importance of adjusting the internal state updates based on observations has also been discussed in recent SSM-based approaches (Gu & Dao, 2023).

## 2.4 DBF FOR NONLINEAR DYNAMICS, LINEAR/NONLINEAR OBSERVATIONS

In this scenario, the Gaussianity of the test distribution is lost during the predict step, making it impossible to apply the original dynamics over the physical variables $z_t$. Therefore, we introduce a new set of latent variables $h_t$ and assume a dynamics model over $h_t$: $p(h_{t+1}|h_t) = \mathcal{N}(h_{t+1}|Ah_t, Q)$ (see panel (b) in Fig. 1). The IOO maps observations into the latent variables $h_t$: $r(h_t|o_t)$. The recursive formula follows Equations 4 and 5. To retrieve the distribution of the original physical variables $z_t$, we introduce an emission model $p(z_t|h_t) = \mathcal{N}(z_t; \phi(h_t), R)$, where $\phi$ is represented by a NN. By marginalizing over $h_t$ with this emission model, a trained DBF can generate samples of $z_t$ that follow the test distribution $q(z_t|o_{1:t})$ given observations $o_{1:t}$.

Although the dynamics operator $A$ for the latent variables $h_t$ is linear, it can express any nonlinear dynamics if the latent space is sufficiently high-dimensional. The Koopman operator (Koopman, 1931) provides a framework for representing nonlinear systems by mapping observables—functions of the system's state—into a higher-dimensional space where the dynamics are linear. For a system $z_{t+1} = f(z_t)$, the Koopman operator $\mathcal{K}$ is a linear operator acting on a set of observables $g(z)$, such that $\mathcal{K}g(z_t) = g(f(z_t))$. This reformulates the system as $h_{t+1} = Ah_t$ in the latent space, where $A$ is the dynamics matrix learned by DBF. While the physical dynamics $f(z)$ are nonlinear, the Koopman operator ensures the existence of an embedding that linearizes the dynamics, enabling recursive computation of test distributions. Discovering such embeddings in finite dimensions has been widely studied (Takeishi et al., 2017; Lusch et al., 2018; Azencot et al., 2020). In high-dimensional simulations, the true degrees of freedom are often far fewer than the simulated variables, making surrogate modeling with the Koopman operator a promising approach to reducing computational costs.

## 2.5 TRAINING

When assimilating in the physical space (i.e., when the dynamics are linear), we train the IOO (i.e., $f_\theta$ and $G_\theta$) by optimizing the evidence lower bound (ELBO) without using the teacher signal $z_t$:

$$\log p(o_{1:T}) = \sum_{t=1}^{T} \log p(o_t|o_{1:t-1}) \geq -\mathcal{L}_{\text{ELBO}},$$

$$\mathcal{L}_{\text{ELBO}} = -\sum_{t=1}^{T} \int q(h_t|o_{1:t}) \log p(o_t|h_t) dh_t + KL[q(h_t|o_{1:t})||q(h_t|o_{1:t-1})], \quad (6)$$

where $KL[p||q]$ denotes the Kullback-Leibler divergence between distributions $p$ and $q$ (see Sec. A.1 in the appendix for the derivation). Here, $q(h_1|o_{1:0}) = q(h_1)$ is the initial distribution. If the SSM contains any unknown parameters, we can train these parameters as well.

For SSMs with nonlinear or unknown dynamics, we have two approaches:

**Strategy 1** Pretrain the Koopman operator, which consists of the nonlinear mapping from $z_t$ to $h_t$, the linear dynamics between $h_t$ and $h_{t+1}$ represented by matrix $A$, and the reverse nonlinear mapping from $h_t$ to $z_t$ denoted by $\phi$. With these components ($A$ and $\phi$) of the Koopman operator, the method designed for linear dynamics can be applied. For pretraining, we require samples of $z_t$ or the SSM for the physical variables to generate these samples. Pairs of $z_t$ and $o_t$ are not necessary, as the training for the linear dynamics ($A$ and $\phi$) and the IOO ($r(h_t|o_t)$) can be performed separately.

**Strategy 2** Train all components (the matrix $A$, the stochastic mapping $p(z_t|h_t) = \mathcal{N}(z_t; \phi(h_t), \text{diag}[\sigma^2])$, and the IOO) simultaneously. In this case, samples of $(z_t, o_t)$ pairs or the SSM for both physical and observation variables to generate these sample pairs are required during training. Note that the physical variables $z_t$ are not required for inference, ensuring that real-time applications are not hindered by the need for $z_t$ during training. The parameters are optimized by maximizing a joint ELBO, $\mathcal{L}_{\text{ELBO,joint}}$, via supervised training:

$$\log p(o_{1:T}, z_{1:T}) = \sum_{t=1}^{T} \log p(o_t, z_t|o_{1:t-1}, z_{1:t-1}) \geq -\mathcal{L}_{\text{ELBO,joint}},$$

$$\mathcal{L}_{\text{ELBO,joint}} = -\sum_{t} \int q(h_t|o_{1:t}) \log p(z_t|h_t) dh_t + KL[q(h_t|o_{1:t})||q(h_t|o_{1:t-1})]. \quad (7)$$

(See Sec. A.2 in the appendix for the derivation). We have replaced $q(h_t|o_{1:t}, z_{1:t})$ with its special case $q(h_t|o_{1:t})$ as our objective is to give the best estimate of $z_t$ given observations $o_{1:t}$.

## 2.6 RELATED WORKS

### 2.6.1 DYNAMICAL VARIATIONAL AUTOENCODERS

DVAEs (see Girin et al. 2021 for a review) are a broad class of models incorporating time-series architectures into VAEs, with DBF as a specialized subcategory. Key differences include (i) the posterior design and realization of the dynamics step, and (ii) the loss function.

**posterior design** Our strategy for the test distribution is to incorporate an appropriate architecture that reflects the Markov property in the time dimension of the test distribution. The IOO, $r(h_t|o_t)$, and the linear dynamics model serve as key instruments in constructing the test posterior distributions. A distinguishing feature of our methodology is that each component's role is defined with respect to the Markov property of the state-space model (SSM) and is clearly differentiated from other components involved in posterior construction. For example, the IOO influences only the update step and does not affect the prediction step. We refer to this methodology as "Bayes-Faithful" due to its tailored design for SSMs that exhibit the Markov property.

In contrast, the test posterior distributions in DVAEs are constructed using RNNs. The complexity of the transition model prevents the analytical computation of latent variables across time steps. As a result, these values can only be estimated via Monte Carlo sampling. Consequently, during inference, successive Monte Carlo sampling ("cascade trick"; Girin et al. 2021) becomes unavoidable.

**loss function** DBF takes the ELBO from factorized density $\log p(o_t|o_{1:t-1})$ in $\log p(o_{1:T}) = \sum_t \log p(o_t|o_{1:t-1})$:

$$\log p(o_{1:T}) \geq \sum_{t=1}^{T} (E_{q(h_t|o_{1:t})}[\log p(o_t|h_t)] - KL[q(h_t|o_{1:t})|q(h_t|o_{1:t-1})]). \tag{8}$$

On the other hand, DVAEs take the ELBO from probability density with all the observations at once.

$$\log p(o_{1:T}) \geq E_{q(h_{1:T}|o_{1:T})}[\log p(h_{1:T}, o_{1:T}) - \log q(h_{1:T}|o_{1:T})]. \tag{9}$$

Therefore, DBF seeks for the filtered distributions $q(h_t|o_{1:t})$ whereas DVAEs model the smoother distributions $q(h_t|o_{1:T})$. Again, for DVAEs, to evaluate the expected values in Equation 9, we need to undergo successive Monte-Carlo sampling over $T$ variables ($h_{1:T}$) (see also Sec. A.3).

Assuming linear Gaussian dynamics and a Gaussian IOO, DBF allows for the analytical integration of $q(h_t|o_{1:t-1})$, resulting in a structured encoder. This structured posterior enables the recursive computation of the filtered distribution $q(h_t|o_{1:t})$ without relying on Monte Carlo sampling, setting it apart from other DVAEs. By constraining the dynamics to be linear, DBF ensures exact integration without the accumulation of Monte Carlo sampling errors across time steps.

Moreover, the linear assumption helps DBF mitigate the instability issues commonly faced when training standard RNNs. The linearity of the latent dynamics is also assumed in normalizing Kalman Filter (de Bézenac et al., 2020) and Kalman variational auto-encoder (Fraccaro et al., 2017). SSMs are increasingly favored for modeling long-range dependencies (Gu & Dao, 2023). S4 (Gu et al., 2022) learns linear dynamics in the latent space, proposing an efficient computation algorithm that outperforms transformers on datasets with long-range dependencies. LS4 (Zhou et al., 2023) extends S4 by introducing stochasticity through a VAE-like structure. Both LS4 and DBF employ linear SSMs and Gaussian posterior approximations, but DBF updates the mean and covariance using a recursive formula based on Bayes' rule, while the construction of posteriors in LS4 is not recursive.

### 2.6.2 KF-BASED METHODS

Various approaches have been explored to address LGSS limitations, including linearizing the model via first-order approximations like the extended Kalman Filter (EKF), approximating populations with a Gaussian distribution in the ensemble Kalman Filter (EnKF; Evensen 1994), and using NNs to approximate the Kalman gain (Revach et al., 2022). EKFNet (Xu & Niu, 2024) assumes EKF for

the construction of test distribution and train the SSM parameters. Auto-EnKF (Chen et al., 2022; 2023) leverages EnKF and train the model by optimizing ELBO. The EnKF and its variants (e.g., ETKF; Bishop et al. 2001) are commonly used in real-time data assimilation for weather forecasting. However, these methods rely on the KF's posterior update equations, limiting the expressivity of the distributions. Additionally, computations for covariance matrices become challenging in high-dimensional spaces, requiring specialized techniques for computational efficiency.

### 2.6.3 SAMPLING-BASED METHODS

The Particle Filter is a popular method for assimilating any posterior. However, achieving adequate particle density in high-dimensional state spaces poses significant challenges. Insufficient density of particles leads to particle degeneracy, where few particles explain the observed data (Beskos et al., 2014). In contrast, DBF directly learns to position density through the IOO, offering advantages for high-dimensional tasks. The Particle Flow Filter (PFF; Daum & Huang 2007; Hu & van Leeuwen 2021) addresses particle degeneracy by moving particles according to gradient flow and effectively scales to nonlinear SSMs with hidden state dimensions up to 1000 (Hu & van Leeuwen, 2021).

### 2.6.4 APPROXIMATE MAP ESTIMATION METHOD

MAP estimation is used to identify the high-density point of the posterior in high-dimensional space, such as in weather forecasting (Lorenc, 2003; Frerix et al., 2021). Even if the computation of the posterior $p(h_t|o_{1:t})$ is intractable, we can optimize $\log p(h_t|o_{1:t}) = \log p(o_t|h_t) + \log p(h_t|o_{1:t-1})$ if we can describe $p(o_t|h_t)$ and $p(h_t|o_{1:t-1}) = \int p(h_t|h_{t-1})p(h_{t-1}|o_{1:t-1})dh_{t-1}$ explicitly. In practice, we cannot access $p(h_{t-1}|o_{1:t-1})$ and therefore the integral $\int p(h_t|h_{t-1})p(h_{t-1}|o_{1:t-1})dh_{t-1}$, so we only compute the mean. The downside is that sequential computation of the covariance matrix of $p(h_t|o_{1:t-1})$ is impossible.

### 2.6.5 NN-BASED PDE SURROGATE

Recently, there have been attempts to approximate partial differential equations (PDEs) using NNs. In this study, we experimented with one of the latest methods, PDE-refiner (Lippe et al., 2023), but its performance was poor and was excluded from the experiments. We suspect this is because PDE-refiner, designed for constructing PDE surrogates, does not handle noisy observations well, making it sensitive to noise. However, we confirmed that it performs well under noiseless conditions.

## 3 EXPERIMENTS

We evaluate the performance of DBF on three tasks: a linear dynamics problem (object tracking) and two nonlinear dynamics problems (double pendulum and Lorenz96). An additional experiment on linear dynamics (moving MNIST) is presented in Sec. B of the appendix.

**Linear dynamics: object tracking** Linear dynamics are applicable in real-world object tracking tasks involving objects with continuous motion, either stationary or nearly uniform linear. An object tracker can be easily constructed by replacing the IOO with an object detector that performs independent detection for each frame. DBF adaptively weights the detected object positions based on the confidence in the current detector's estimates, significantly enhancing tracking robustness. The results are compared against those of the KF.

**Nonlinear dynamics: double pendulum and Lorenz96** For nonlinear dynamics problems, such as the double pendulum and Lorenz96, DBF constructs a new latent space in addition to the original physical space. Here, we took Strategy 2 in Sec.2.5 for the training: we simultaneously train NNs for the IOO, nonlinear observation operator $\phi$, the dynamics matrix $A$, and the emission model's standard deviation. We compare the performance of DBF with the classical DA algorithms (EnKF, ETKF, PF), state-of-the-art assimilation methodologies (PFF Daum & Huang 2007; Hu & van Leeuwen 2021, KalmanNet Revach et al. 2022), and DVAE-based approaches (deep Kalman Filter; DKF, Krishnan et al. 2015; 2016, variational recurrent neural network; VRNN, Chung et al. 2015, and stochastic recurrent neural network; SRNN, Fraccaro et al. 2016). DBF and other DVAEs are trained by optimizing the evidence lower bound (ELBO), as described in Sec. 2.5.

For nonlinear dynamics experiments, we generate random initial conditions and evolve them using the dynamics. Synthetic observations are produced by applying the observation operator with additive noise. Noise levels, observation operators, and further details are given in Sec. C, C.1, C.2, and C.3. Sec. C also provides computationally efficient parametrization of the latent dynamics matrix.

### 3.1 LINEAR DYNAMICS: OBJECT TRACKING

In a single-object tracking problem, a detector identifies a bounding box for the object in each frame, and these boxes are then connected across frames. When the object is not fully visible or is obscured, the detector often fails to accurately determine its position. In such scenarios, the KF aids by predicting and assimilating the object's true position. However, a key limitation of the KF is its reliance on a fixed observation model throughout the tracking process. While low-confidence observations can provide valuable approximate position information, they may also mislead the tracker with inaccurate data, potentially degrading overall tracking performance.

We demonstrate that DBF can enhance tracking stability without requiring additional training. During the computation of the posterior $p(z_t|o_{1:t})$ from $p(z_t|o_{1:t-1})$, the importance of the observation $o_t$ is regulated via $G_\theta(o_t)$. This allows the observational confidence to be effectively incorporated into the posterior estimation. We evaluate the tracking performance using the "airplane" category from the LaSOT dataset (Fan et al., 2019; 2021).

We use the first 1,000 frames from 20 videos for evaluation. The first 10 videos serve as a validation set for determining filter parameters (see Sec. C.1), while the performance is assessed using videos 11–20. Each set of 1,000 frames is divided into 20 subsets of 50 frames. Filters are initialized at the ground truth coordinates of the bounding box in the first frame, after which each filter is responsible for tracking the bounding box throughout the subset. We employ the YOLOv8n model (Jocher et al., 2023) as the object detector. The detector outputs the bounding box position, $X$, along with a confidence score, $c$. A detection threshold of 0.01 is applied. When multiple bounding boxes are detected, the one with the highest posterior probability is selected.

The bounding box coordinates are used as $f_\theta(o_t) = X$. We experiment with linear confidence $G_\theta(o_t) \propto c$ and squared confidence $G_\theta(o_t) \propto c^2$ and find that the squared confidence $G_\theta(o_t) \propto c^2$ perform better. For further settings, see Sec. C.1.

Figure 2 presents the results. The left panel provides an illustrative example comparing the two tracking algorithms. The KF tracker is visibly influenced by false detections, being pulled toward a coordinate value of approximately 150 during frames 15–17. In contrast, the DBF tracker maintains stable predictions under the same conditions. The middle and the right panels offer a quantitative comparison of KF and DBF in terms of intersection over union (IoU). Both filters perform well in estimating bounding box positions in frames without detections. However, DBF demonstrates a significant performance advantage in frames with low-confidence ($c < 0.1$) detections. This improvement can be attributed to DBF's flexibility, allowing it to adaptively decide whether to trust low-confidence observations or disregard them.

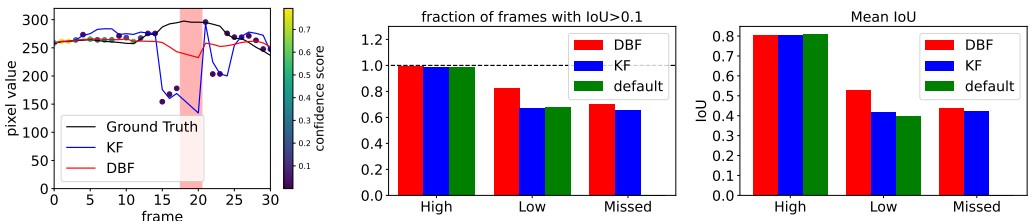

Figure 2: Left panel: x coordinate of bounding box center estimated with KF and DBF. Colored dots show the coordinates of the bounding box reported by the YOLO model. The red band (frames 18 − 20) shows frames where the detection network reports no bounding boxes. Middle panel: fraction of frames with IoU > 0.1 for each tracker. Detections with a confidence score greater than 0.1 are categorized as "High", those below 0.1 as "Low", and those below 0.01 as "Missed". Right panel: mean IoU for the three categories. The performance gain of DBF from KF is considerable in frames with low-confidence detections.

## 3.2 Nonlinear dynamics 1: double pendulum

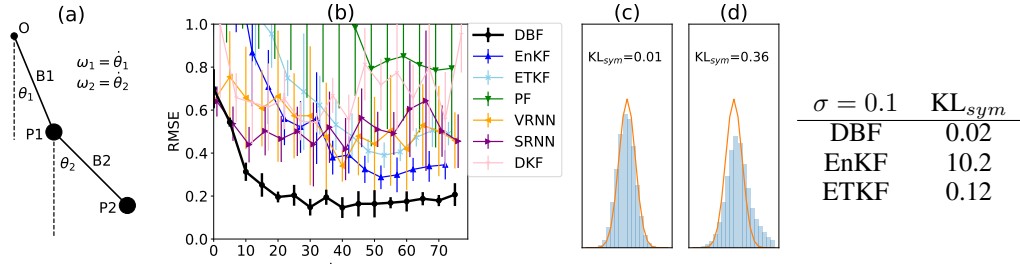

Figure 3: A schematic figure (panel a) and results for double pendulum experiments. Panel (b) shows the RMSE of angle velocities (averaged over $\omega_1$ and $\omega_2$) over time steps. Panels (c) and (d) show example histograms for normalized errors in DBF and ETKF samples compared against the unit Gaussian $\mathcal{N}(x; \mu = 0, \sigma^2 = 1)$. The small table compares the Jeffreys divergence of normalized errors and the unit Gaussian between DBF, EnKF, and ETKF predictions.

Table 1: RMSE at the final ten steps of assimilation in double pendulum experiments.

|  | $\sigma = 0.1$ | | $\sigma = 0.3$ | | $\sigma = 0.5$ | |
|---|---|---|---|---|---|---|
|  | $\theta$ | $\omega$ | $\theta$ | $\omega$ | $\theta$ | $\omega$ |
| DBF | $\mathbf{0.03 \pm 0.01}$ | $\mathbf{0.21 \pm 0.04}$ | $\mathbf{0.05 \pm 0.02}$ | $\mathbf{0.26 \pm 0.05}$ | $\mathbf{0.06 \pm 0.01}$ | $\mathbf{0.36 \pm 0.04}$ |
| EnKF | $0.05 \pm 0.00$ | $0.33 \pm 0.07$ | $0.14 \pm 0.01$ | $0.71 \pm 0.09$ | $0.24 \pm 0.01$ | $1.17 \pm 0.22$ |
| ETKF | $0.05 \pm 0.01$ | $0.46 \pm 0.08$ | $0.22 \pm 0.05$ | $1.41 \pm 0.41$ | $0.36 \pm 0.08$ | $2.70 \pm 1.25$ |
| PF | $0.05 \pm 0.00$ | $0.63 \pm 0.24$ | $0.21 \pm 0.14$ | $1.41 \pm 1.30$ | $0.32 \pm 0.08$ | $2.36 \pm 2.29$ |
| PFF | $1.27 \pm 0.29$ | $1.04 \pm 0.15$ | NA | $5.99 \pm 1.09$ | $5.88 \pm 0.67$ | NA |
| KNet | NA | NA | NA | NA | NA | NA |
| VRNN | $0.04 \pm 0.01$ | $0.44 \pm 0.19$ | $0.06 \pm 0.02$ | $0.35 \pm 0.14$ | $0.08 \pm 0.04$ | $0.40 \pm 0.16$ |
| SRNN | $0.05 \pm 0.02$ | $0.52 \pm 0.18$ | $0.06 \pm 0.02$ | $0.44 \pm 0.08$ | $0.08 \pm 0.03$ | $0.52 \pm 0.22$ |
| DKF | $0.12 \pm 0.02$ | $2.70 \pm 0.28$ | $0.17 \pm 0.03$ | $2.61 \pm 0.74$ | $0.23 \pm 0.04$ | $2.61 \pm 0.56$ |

This section presents our experiments with a double pendulum system, selected for its nonlinear and chaotic behavior. The pendulum consists of two 1 kg masses, P1 and P2, connected by two 1 meter bars, B1 and B2. One end of the bar B1 is fixed at the origin ("O"), with the other end attached to P1. Mass P2 is connected to P1 via bar B2. A schematic of the setup is shown in panel (a) of Fig. 3.

We use the angles $\theta_1$ and $\theta_2$, and the two angular velocities, $\omega_1$ and $\omega_2$, as target physical variables. The latent dimension for DBF, VRNN, SRNN, and DKF is set to 50. For the choice of the latent dimensions, refer to Sec. E.1. Observation data consists of the two-dimensional spatial positions of masses $P_1$ and $P_2$, corrupted by Gaussian noise. The observation operator combines trigonometric functions for $\theta_1$ and $\theta_2$ which are highly nonlinear. Experiments are conducted with noise levels of $\sigma = 0.1, 0.3$, and 0.5 [m], with a time step of 0.03 [s] between observations. In the emission model $p(z_t|h_t)$, we assume von Mises distributions for $\theta_1$ and $\theta_2$, while $\omega_1$ and $\omega_2$ follow Gaussians.

Table 1 presents the RMSE between the physical variables and the mean of the filtered distribution. For both the angles $\theta$ and angle velocities $\omega$, we compute the averages of the two variables across two pendulums. Training for KalmanNet was unsuccessful under all conditions. For the DVAEs, we exclude failed initial conditions (2/15 for VRNN and DKF, and 3/15 for SRNN) when calculating the RMSE. DBF outperforms both model-based and latent assimilation methods across all settings, showing significant improvements in estimating $\omega$, which cannot be inferred from a single observation. Fig. 3 (b) illustrates an example of RMSE evolution during assimilation, where DBF consistently outperforms the other methods. The assimilation of $\omega$ occurs within the first $\sim 20$ steps, maintaining an excellent estimation accuracy throughout the experiment.

A key feature of DBF is its ability to generate samples of $z_t$ and assess the uncertainty in state estimates. To evaluate this capability, we analyze the distributions of normalized errors defined as $\epsilon_{norm,t,i} = (z_{t,sample,i} - z_{t,i})/\delta_i$, where $z_{t,i}$ represents the true value of dimension $i$ at time $t$, and $\delta_i$ is the standard deviation of $z_{t,sample,i}$. We collect $\epsilon_{norm,t,i}$ across all time steps, focusing

on $i = \omega_1$ and $i = \omega_2$, since $\theta_1$ and $\theta_2$ follow von Mises distributions. If the uncertainty estimates are accurate, $\epsilon_{norm,t,i}$ should approximate a Gaussian distribution with a standard deviation of one. To quantify the accuracy, we compute the symmetric KL divergence (Jeffreys divergence) $KL_{sym}[p,q] = (KL[p||q] + KL[q||p])/2$ between the histogram of $\epsilon_{norm,t,i}$ and a unit Gaussian. DBF exhibits very low $KL_{sym}$ values, indicating accurate error estimation. Panels (c) and (d) display example histograms of $\epsilon_{norm,t,i}$ for DBF and ETKF.

### 3.3 NONLINEAR DYNAMICS 2: LORENZ96

In the final experiment, we focus on state estimation in the Lorenz96 model (Lorenz, 1995), a benchmark for testing data assimilation algorithms on noisy, nonlinear observations. The Lorenz96 model describes the evolution of a one-dimensional array of variables, each representing a physical quantity over a spatial domain, like an equilatitude circle. The dynamics are governed by the following coupled ordinary differential equations:

$$\frac{dz_i}{dt} = (z_{i+1} - z_{i-2})z_{i-1} - z_i + F, \quad i = 1, \dots, N, \quad (10)$$

where $z_i$ is the value at grid $i$, $N$ is the number of grid points, and $F$ is external forcing. For our experiments, we take $(F, N) = (8, 40)$.

We consider two observation operators. The first adds Gaussian noise to direct observations: $o_{t,j} = z_{t,j} + \epsilon$, with noise levels $\sigma = 1, 3, 5$. The second uses a nonlinear operator: $o_{t,j} = \min(z_{t,j}^4, 10) + \epsilon$, with the same noise levels. The dynamic range of $z_{t,j}$ is around $\pm 10$, and observations are capped at 10 when $z_{t,j}$ exceeds 1.8. This makes it highly challenging for classical DA

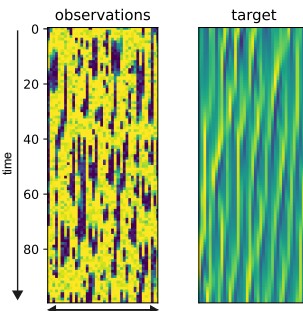

Figure 4: A Hovmöller diagram for one of data in the test set. The observation operator is nonlinear, $o_{t,j} = min(z_{t,j}^4, 10) + \epsilon$.

methods, as each observation offers limited information. The filter must integrate data over long timesteps, where nonlinear dynamics distort the probability distribution. Fig. 4 illustrates observations and target values. All models use 80 observation steps with a 0.03 time interval. The latent dimension for DBF, VRNN, SRNN, and DKF is set to 800 (for the choice of the latent dimension in DBF, see Sec. E.2). For further details for the experiment, see Sec. C.3.

Table 2: RMSE at the final ten steps of assimilation in Lorenz96 experiments.

| | direct observation | | | nonlinear observation | | |
|---|---|---|---|---|---|---|
| | $\sigma = 1$ | $\sigma = 3$ | $\sigma = 5$ | $\sigma = 1$ | $\sigma = 3$ | $\sigma = 5$ |
| DBF | $0.53 \pm 0.04$ | $\mathbf{0.82 \pm 0.03}$ | $\mathbf{1.16 \pm 0.07}$ | $\mathbf{1.08 \pm 0.15}$ | $\mathbf{1.29 \pm 0.18}$ | $\mathbf{1.65 \pm 0.17}$ |
| EnKF | $0.31 \pm 0.01$ | $0.83 \pm 0.10$ | $1.73 \pm 0.12$ | $4.69 \pm 0.14$ | $3.93 \pm 0.08$ | $3.81 \pm 0.07$ |
| ETKF | $\mathbf{0.30 \pm 0.01}$ | $1.06 \pm 0.15$ | $2.42 \pm 0.11$ | $4.57 \pm 0.25$ | $4.28 \pm 0.04$ | $4.23 \pm 0.07$ |
| PF | $2.80 \pm 0.04$ | $3.12 \pm 0.06$ | $3.62 \pm 0.13$ | $6.05 \pm 0.16$ | $4.95 \pm 0.12$ | $4.58 \pm 0.14$ |
| PFF | $0.60 \pm 0.02$ | $1.00 \pm 0.05$ | $2.20 \pm 0.09$ | $3.75 \pm 0.09$ | $3.85 \pm 0.04$ | $3.83 \pm 0.11$ |
| KNet | $0.60 \pm 0.02$ | $1.81 \pm 0.05$ | $3.02 \pm 0.09$ | $2.97 \pm 0.21$ | $3.47 \pm 0.17$ | $3.99 \pm 0.25$ |
| VRNN | $3.67 \pm 0.06$ | $3.67 \pm 0.06$ | $3.67 \pm 0.06$ | $3.69 \pm 0.04$ | $2.51 \pm 0.79$ | $3.67 \pm 0.06$ |
| SRNN | $3.08 \pm 0.56$ | $3.63 \pm 0.05$ | $3.40 \pm 0.29$ | $3.30 \pm 0.81$ | $3.62 \pm 0.41$ | $2.96 \pm 0.32$ |
| DKF | $3.70$ | NA | NA | NA | NA | NA |

Table 2 presents the assimilation performance across different noise levels and observation settings. DBF outperforms existing methods in direct observations with $\sigma = 3, 5$, and across all noise levels for nonlinear observation cases. In the $\sigma = 1$ setting with direct observation, traditional algorithms like EnKF and ETKF outperform DBF.

The superior performance of EnKF and ETKF with direct observations at the lowest noise level can be attributed to the minimal non-Gaussianity in the posteriors within physical space. Non-Gaussianity can originate from both the dynamics model (predict step) and the observation model (update step). In this setting, the linearity of the observation operator prevents non-Gaussianity from being introduced during the update step, provided that the prior $q(z_t|o_{1:t-1})$ is Gaussian. Additionally, state estimation from each observation is highly accurate due to small noise. As a result, the prior $q(z_t|o_{1:t-1})$ remains close to a Gaussian distribution, as the locally linear approximation of

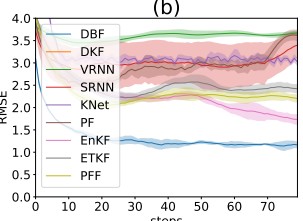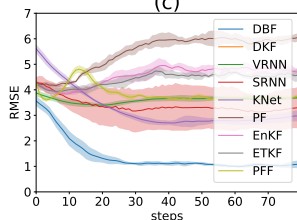

Figure 5: RMSE results for Lorenz96 experiments. Panels (a), (b) show results for direct observation with $\sigma = 1$ and $\sigma = 5$. Panel (c) shows results for nonlinear observation with $\sigma = 1$.

the dynamics adequately captures the time evolution of probability distributions. The poorer performance of EnKF and ETKF in the $\sigma = 5$ experiment is attributed to the increased non-Gaussianity introduced during each predict step. Similarly, when the observation operator is nonlinear, each update step introduces substantial non-Gaussianity. This results in a significant drop in performance for traditional filtering methods across all noise levels. In these scenarios, DBF consistently maintains an advantage over classical DA algorithms.

We observe that training DVAE-based methods is highly unstable, while that for DBF exhibits stability. Dynamics in DVAEs are modeled by RNNs, which often suffer from unstable training due to exploding or vanishing gradients. In contrast, DBF employs matrix multiplication for dynamics. If the eigenvalues of the matrix exceed one by a large margin, the model predictions, and consequently the loss function, would explode irrespective of inputs. Fig. 6 shows the histogram of the absolute values of eigenvalues at the end of training, which are distributed around or below one, indicating stable training.

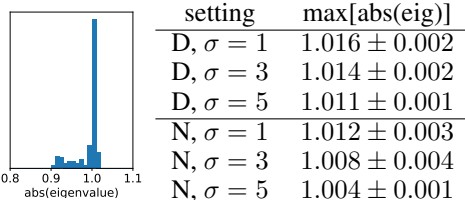

| setting | max[abs(eig)] |
|---|---|
| D, $\sigma = 1$ | $1.016 \pm 0.002$ |
| D, $\sigma = 3$ | $1.014 \pm 0.002$ |
| D, $\sigma = 5$ | $1.011 \pm 0.001$ |
| N, $\sigma = 1$ | $1.012 \pm 0.003$ |
| N, $\sigma = 3$ | $1.008 \pm 0.004$ |
| N, $\sigma = 5$ | $1.004 \pm 0.001$ |

Figure 6: Histogram of 800 eigenvalues of the dynamics matrix in Lorenz96. D for direct and N for nonlinear observations.

## 4 LIMITATION

DBF's learning of IOO requires a training phase, unlike classical model-based data assimilation methods. Specifically, when dealing with nonlinear dynamics, DBF requires either: (i) a pair of $(z_t, o_t)$ generated from the original SSM, (ii) a pair of $(z_t, o_t)$ obtained via, e.g., retrospective reanalysis (ERA5; Hersbach et al. 2020 in weather forecasting), or (iii) a pretrained Koopman operator and observed data $o_t$.

In the Lorenz96 experiment, DBF's performance with direct observation with $\sigma = 1$ falls short compared to EnKF and ETKF. In this setting, the non-Gaussianity of posteriors is weak, resulting in minor approximation errors due to Gaussian assumptions. Consequently, a model-based approach may be more advantageous in such situations, as it leverages complete SSM knowledge without introducing training biases.

## 5 CONCLUSION

We propose DBF, a novel DA method. DBF is a NN-based extension of the KF designed to handle nonlinear observations. While constraining the test distributions to remain Gaussian, DBF enhances their representational capacity by leveraging nonlinear transform expressed by a NN. DBF is the first "Bayes-Faithful" amortized variational inference methodology, constructing test distributions that mirror the inference structure of a SSM with the Markov property. This structured inference enables analytical computation of test distributions, preventing the accumulation of Monte Carlo sampling errors over time steps. DBF exhibits superior performance over existing methods in scenarios where posterior distributions become highly non-Gaussian, such as in the presence of nonlinear observation operators or significant observation noise.

**Reproducibility Statement** We have provided the source code to reproduce the experiments for double pendulum (Sec. 3.2) and the Lorenz96 (Sec. 3.3) in the supplementary material. The hyperparameters for the training are provided in Table 3 in the appendix. Generation method of the training and test dataset, the dynamics model, the observation model, and the architectures are detailed in the appendix: Sec. C.1, C.2, and C.3.

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

# A DERIVATION OF THE EVIDENCE LOWER-BOUND AND THE ASSOCIATED MONTE-CARLO SAMPLING

## A.1 LINEAR DYNAMICS CASE

Following the definition of the probability density,

$$p(o_t, h_t|o_{1:t-1}) = p(o_t|o_{1:t-1})p(h_t|o_{1:t}) \tag{11}$$

Using Eq. 11 at the third equality,

$$
\begin{aligned}
\log p(o_{1:T}) &= \sum_{t=1}^{T} \log p(o_t|o_{1:t-1}) \\
&= \sum_{t=1}^{T} \int q(h_t|o_{1:t}) \log p(o_t|o_{1:t-1}) dh_t \\
&= \sum_{t=1}^{T} \int q(h_t|o_{1:t}) \log \frac{p(o_t, h_t|o_{1:t-1})}{p(h_t|o_{1:t})} dh_t \\
&= \sum_{t=1}^{T} \int q(h_t|o_{1:t}) \log \left[ \frac{p(o_t, h_t|o_{1:t-1})}{q(h_t|o_{1:t})} \frac{q(h_t|o_{1:t})}{p(h_t|o_{1:t})} \right] dh_t \\
&= \sum_{t=1}^{T} \int q(h_t|o_{1:t}) \log \left[ \frac{p(o_t, h_t|o_{1:t-1})}{q(h_t|o_{1:t})} \right] dh_t + KL[q(h_t|o_{1:t})||p(h_t|o_{1:t})] \\
&= \sum_{t=1}^{T} \mathcal{L}_{ELBO,t} + KL[q(h_t|o_{1:t})||p(h_t|o_{1:t})] \\
&\geq \sum_{t=1}^{T} \mathcal{L}_{ELBO,t} \quad (12) \\
\mathcal{L}_{ELBO,t} &= \int q(h_t|o_{1:t}) \log \left[ \frac{p(o_t, h_t|o_{1:t-1})}{q(h_t|o_{1:t})} \right] dh_t \\
&= \int q(h_t|o_{1:t}) \log \left[ \frac{p(h_t|o_{1:t-1})p(o_t|h_t)}{q(h_t|o_{1:t})} \right] dh_t \\
&= \int q(h_t|o_{1:t}) \log p(o_t|h_t) dh_t + \int q(h_t|o_{1:t}) \frac{p(h_t|o_{1:t-1})}{q(h_t|o_{1:t})} dh_t \\
&= \int q(h_t|o_{1:t}) \log p(o_t|h_t) dh_t - KL[q(h_t|o_{1:t})|p(h_t|o_{1:t-1})] \quad (13)
\end{aligned}
$$

The true prior at step $t$ ($p(h_t|o_{1:t-1})$) on the right hand side of Eq. 13 could be replaced with the prior computed from the test distribution $q(h_t|o_{1:t-1})$ when training.

## A.2 NONLINEAR DYNAMICS CASE

$$
p(o_t, z_t, h_t|o_{1:t-1}, z_{1:t-1}) = p(o_t, z_t|o_{1:t-1}, z_{1:t-1})p(h_t|o_{1:t}, z_{1:t}) \quad (14)
$$

The derivation proceeds parallel to the linear case. Using Eq. 14 at the third equality,

$$
\begin{aligned}
\log p(o_{1:T}, z_{1:T}) &= \sum_{t=1}^{T} \log p(o_t, z_t | o_{1:t-1}, z_{1:t-1}) \\
&= \sum_{t=1}^{T} \int q(h_t|o_{1:t}) \log p(o_t, z_t | o_{1:t-1}, z_{1:t-1}) dh_t \\
&= \sum_{t=1}^{T} \int q(h_t|o_{1:t}) \log \frac{p(o_t, z_t, h_t | o_{1:t-1}, z_{1:t-1})}{p(h_t|o_{1:t}, z_{1:t})} dh_t \\
&= \sum_{t=1}^{T} \int q(h_t|o_{1:t}) \log \left[ \frac{p(o_t, z_t, h_t | o_{1:t-1}, z_{1:t-1})}{q(h_t|o_{1:t})} \frac{q(h_t|o_{1:t})}{p(h_t|o_{1:t}, z_{1:t})} \right] dh_t \\
&= \sum_{t=1}^{T} \int q(h_t|o_{1:t}) \log \left[ \frac{p(o_t, z_t, h_t | o_{1:t-1}, z_{1:t-1})}{q(h_t|o_{1:t})} \right] dh_t + KL[q(h_t|o_{1:t})||p(h_t|o_{1:t}, z_{1:t})] \\
&= \sum_{t=1}^{T} \mathcal{L}_{ELBO,joint,t} + KL[q(h_t|o_{1:t})||p(h_t|o_{1:t}, z_{1:t})] \\
&\geq \sum_{t=1}^{T} \mathcal{L}_{ELBO,joint,t}
\end{aligned}
\tag{15}
$$

$$
\begin{aligned}
\mathcal{L}_{ELBO,joint,t} &= \int q(h_t|o_{1:t}) \log \left[ \frac{p(o_t, z_t, h_t | o_{1:t-1}, z_{1:t-1})}{q(h_t|o_{1:t})} \right] dh_t \\
&= \int q(h_t|o_{1:t}) \log \left[ \frac{p(h_t|o_{1:t-1}, z_{1:t-1}) p(o_t, z_t | h_t)}{q(h_t|o_{1:t})} \right] dh_t \\
&= \int q(h_t|o_{1:t}) [\log p(z_t|h_t) + \log p(o_t|z_t)] dh_t + \int q(h_t|o_{1:t}) \frac{p(h_t|o_{1:t-1}, z_{1:t-1})}{q(h_t|o_{1:t})} dh_t \\
&= \int q(h_t|o_{1:t}) \log p(z_t|h_t) dh_t - KL[q(h_t|o_{1:t})|p(h_t|o_{1:t-1}, z_{1:t-1})] + \log p(o_t|z_t) \quad (16)
\end{aligned}
$$

The true prior at step $t$ ($p(h_t|o_{1:t-1}, z_{1:t-1})$) on the right hand side of Eq. 16 could be replaced with the prior computed from the test distribution $q(h_t|o_{1:t-1})$ when training. The last term of the equation ($\log p(o_t|z_t)$) can be neglected as it does not affect the new latent variables $h_t$.

### A.3 COMPARISON TO OTHER DVAEs IN TERMS OF MONTE-CARLO SAMPLING

The crucial difference from other DVAEs is that the Monte-Carlo samplings in DBF are not nested with each other. In DVAE, we need to evaluate an integral term $\int q(h_{1:T}|o_{1:T}) \log p(o_{1:T}, h_{1:T}) dh_{1:T}$, where $q(h_{1:T}|o_{1:T}) = \prod_t q(h_t|h_{t-1}, o_t)$. Although the log-term could be factorized as $\sum_t \log p(o_t|h_t) + \log p(h_t|h_{t-1})$ thanks to the Markov property, we need MC (nested) sequential sampling over $h_{1:T}$ if we want to evaluate the term at $t = T$. On the other hand, ELBO in DBF is $\sum_t \int q(h_t|o_{1:t}) \log p(z_t|h_t) dh_t + KL[q(h_t|o_{1:t})|q(h_t|o_{1:t-1})]$ because DBF takes the lower limit of $\sum_t \log p(o_t|o_{1:t-1})$. Thanks to the analytic expressions of $q(h_t|o_{1:t})$ and $q(h_t|o_{1:t-1})$, the KL term can be computed analytically. A MC sampling is needed to compute $\int q(h_t|o_{1:t}) \log p(z_t|h_t) dh_t$ but this is independent from other timesteps.

## B ADDITIONAL LINEAR DYNAMICS EXPERIMENT: TWO-BODY MOVING MNIST

This experiment demonstrates DBF's ability to handle linear dynamics where key parameters of the observation operator are unknown. The dataset consists of 2D figures containing two embedded images, each moving at a constant speed and bouncing off frame edges. The system's physical state

is described by eight variables: the positions $(x, y)$ and velocities $(v_x, v_y)$ of the two embedded images. The dynamics matrix is block-diagonal, composed of four (two-body times two dimensions) translation matrices, $A_{tr}$: $A_{tr} = \begin{pmatrix} 1 & 1 \\ 0 & 1 \end{pmatrix}, z_t = \begin{pmatrix} x_t \\ v_{x_t} \end{pmatrix}$. Observations are corrupted by additive Gaussian noise with a standard deviation of $\sigma = 50$ per pixel, where the original pixel values range from 0 to 255 (see panel (a) of Fig. 7 for an example of the data provided).

The aim is to show that DBF can track the linear dynamics while estimating unknown system parameters. DBF learns the pixel values of the embedded images from noisy observations, while maintaining consistency with physical motion. The observation model contains 1,568 unknown parameters, corresponding to the number of pixels in the images. In classical DA algorithms, it is not possible to train unknown system parameters. However, it may be possible to infer these parameters by incorporating them as new physical dimensions. We have adopted this strategy for classical DA algorithms (EnKF, ETKF, and PF). For these, we tested at three different model noise levels ($\sigma_{sys}$ of 1, 0.1, and 0.01) and chosen the best parameter. While DVAE generates latent variables, they are different from the state variables of the original SSM: therefore, they cannot infer the position or velocity from those images. We were unable to compare with KalmanNet as the high observation dimensions of $x_{dim}^2 = (44 \times 44)^2$ inhibits the training even with the batch size of one.

Fig. 7 summarizes the experiment. Panel (a) shows an example from the test set, illustrating the challenges posed by strong noise and overlapping images. Panel (b) presents the DBF learning process. In the rightmost table, we compare the success rates of DBF against model-based approaches (EnKF, ETKF, PF). We define success as achieving a root-mean-square error (RMSE) of less than 1.0 for both position $(x_1, y_1, x_2, y_2)$ and velocity $(v_{x_1}, v_{y_1}, v_{x_2}, v_{y_2})$ of the two digits over the final ten steps. DBF successfully performs assimilation without explicit knowledge of the images, while all the other model-based approaches fail. The KF-inspired approaches (EnKF, ETKF) failed because of very strong non-Gaussianity in the observation process and the high system dimension. Similarly, PF underperformed because the number of particles (10,000) was insufficient for the problem dimension ($z_{dim} = 8$ and two digits images $2 \times 28 \times 28 = 1,568$). Figures for visualizing the assimilation results for all the algorithms are given in the appendix (Fig. 13).

Panel (b) of Fig. 7 illustrates the evolution of the estimated figures. Initially, DBF assumes two random shapes. "Iterations" in panel (b) means the number of parameter update steps the DBF has undergone. As training progresses, it first identifies one of the numbers ("9") and subsequently detects the second shape ("5"). By the end of the training process, DBF nearly perfectly estimates the parameters of the observation model, including the positions of the figures, which is crucial for adjusting their reflective behavior.

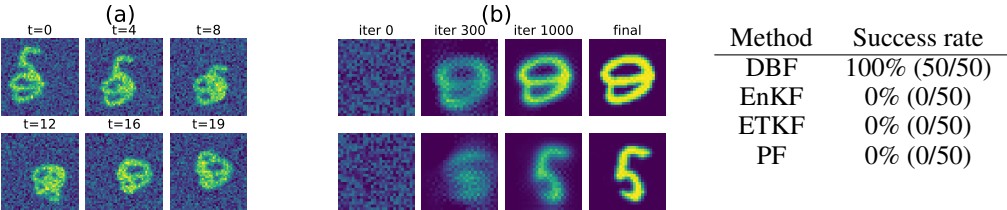

Figure 7: Figures from the two-body Moving MNIST experiments. Panel (a) displays examples of the observation data. Panel (b) illustrates the evolution of the observation model parameters (the embedded images) during training. The table compares the success rates of four methodologies.

## C  SETTINGS AND ADDITIONAL RESULTS FOR EXPERIMENTS

**parametrization of the dynamics matrix**  We have parametrized the dynamics matrix $A$ following Lusch et al. (2018): we consider that $h_{dim}/2$ complex eigenvalues $\lambda_i (0 \le i < h_{dim}/2)$ characterize $A$. Namely, $A$ is a block-diagonal matrix of $h_{dim}/2$ blocks. Each block consists of $2 \times 2$ matrix, whose components are:

$$A_{block} = \exp(\rho_i) \begin{pmatrix} \cos(\omega_i) & -\sin(\omega_i) \\ \sin(\omega_i) & \cos(\omega_i) \end{pmatrix}, \tag{17}$$

where $\rho_i = \text{Re}[\lambda_i]$ and $\omega_i = \text{Im}[\lambda_i]$. In contrast to Lusch et al. (2018), we apply the same dynamics matrix at any positions on the latent space. We consider that this representation is sufficiently expressive, as it can express any matrix on a complex number field that is diagonalizable.

One key advantage of DBF is that augmenting the latent dimension only results in a linear increase in computational demand. This scaling is due to the efficient parametrization of the dynamics matrix, where the block-diagonal structure allows operations to scale linearly with the latent dimension. In contrast, methods such as Sequential Monte Carlo (SMC) suffer from exponential increases in computational demand as the latent space grows, assuming that the same density of particles must be maintained to capture posterior distributions. This makes DBF particularly well-suited for high-dimensional systems where traditional methods struggle with computational complexity.

**Computational resources** We conduct experiments on a cluster of V100 GPUs. Each GPU has memory of 32GB.

**hyperparameters for training** For all experiments, we have used Adam optimizer with default parameters. Table 3 shows hyperparameters employed in our experiments. Trainings for moving MNIST and double pendulum are conducted with one GPU, while that for Lorenz96 is with eight GPUs.

Table 3: Hyperparameters for training

|  | lr | batch size | $h_{dim}$ | $N_{data,train}$ | Epochs | train time per model |
|---|---|---|---|---|---|---|
| object tracking | - | - | 8 | - | - | - |
| double pendulum | $10^{-3}$ | 256 | 50 | $1.0 \times 10^7$ | 1 | 6hr$\times$ 1GPU |
| Lorenz96 | $3 \times 10^{-3}$ | 64 | 800 | $2.6 \times 10^7$ | 1 | 15hr$\times$ 8GPUs |
| moving MNIST | $10^{-3}$ | 64 | 8 | 480,000 | 2 | 3hr$\times$ 1GPU |

## C.1 OBJECT TRACKING

**Dataset:** "Airplane" movies in the LaSOT dataset (Fan et al., 2019; 2021). It contains 20 movies. Each movie has at least 1,000 frames. We chop the first 1,000 frames into 20 sets of 50 frames. Airplanes numbered one to ten are considered a validation set used to determine the model hyperparameters. We use the remaining data (airplane-11 to airplane-20) as a test set to evaluate the performance of the filters.

**Dynamics model:** Constant velocity model. The $(x, y)$ coordinates and $(v_x, v_y)$ velocities of the top left and bottom right edges are the latent (physical) variables.

$$h_{t+1} = Fh_t \tag{18}$$

$$F = \begin{pmatrix} 1 & 0 & 0 & 0 & dt & 0 & 0 & 0 \\ 0 & 1 & 0 & 0 & 0 & dt & 0 & 0 \\ 0 & 0 & 1 & 0 & 0 & 0 & dt & 0 \\ 0 & 0 & 0 & 1 & 0 & 0 & 0 & dt \\ 0 & 0 & 0 & 0 & 1 & 0 & 0 & 0 \\ 0 & 0 & 0 & 0 & 0 & 1 & 0 & 0 \\ 0 & 0 & 0 & 0 & 0 & 0 & 1 & 0 \\ 0 & 0 & 0 & 0 & 0 & 0 & 0 & 1 \end{pmatrix}, h_t = \begin{pmatrix} x_{1,t} \\ y_{1,t} \\ x_{2,t} \\ y_{2,t} \\ v_{x_{1,t}} \\ v_{y_{1,t}} \\ v_{x_{2,t}} \\ v_{y_{2,t}} \end{pmatrix}. \tag{19}$$

Here, $x_{1,t}$ and $y_{1,t}$ stand for the coordinates of the left top edge of the bounding box, and $x_{2,t}$ and $y_{2,t}$ are the right bottom edge of the box. $v_{x_{1,t}}, v_{y_{1,t}}, v_{x_{2,t}}, v_{y_{2,t}}$ are velocities of box edges. $dt$ is the time difference between frames, which we take as 1 (arbitrary).

**Network architecture:** We use a pre-trained detector YOLOv8n model (Jocher et al., 2023). The detector yields the bounding box's position, $X$, and the box's confidence score, $c$. We set the detection threshold at 0.01. In cases where the detector reports multiple bounding boxes, we choose the one with the highest posterior probability. We use the bounding box coordinates as $f_\theta(o_t) = X$. Several choices for the relation between confidence score and $G_\theta(o_t)$ are possible.

We experiment with linear confidence $G_\theta(o_t) \propto c$ and squared confidence $G_\theta(o_t) \propto c^2$. We determine the system noise factor for either dependence with the validation set. We use normalized precision as the evaluation metric (Müller et al., 2018). Figure 8 shows the normalized precision score for the validation set for the system noise factor. The system noise factor of $10^{-1}$ is chosen for KF. For DBF, squared confidence with the system noise factor of $10^{-2}$ is employed.

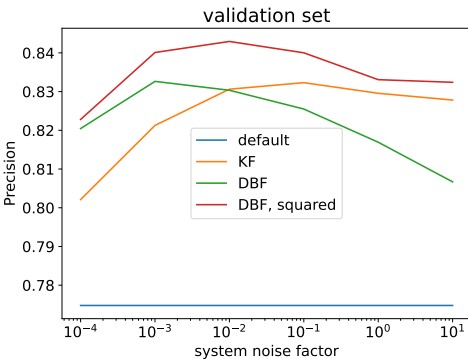

Figure 8: Normalized precision scores for validation samples.

## C.2 DOUBLE PENDULUM

**Dataset:** The dataset consists of 2D coordinates representing the positions of two weights. The training set includes $10,240,000$ initial conditions, while the test set contains 10 initial conditions. The number of training samples is sufficiently large to ensure that the training converges. During DVAE training, we observed that some initial conditions resulted in training failure due to instability; however, we maintained the total number of training samples since the training was successful for at least one initial condition. Both datasets comprise 80 time steps. Numerical integration is performed using the `solve_ivp` function in SciPy, with relative tolerance (`rtol`) set to $10^{-2}$ and absolute tolerance (`atol`) set to $10^{-2}$.

A schematic figure explaining the problem setting is presented in panel (a) of Fig. 3 in the main text.

Dynamics model is described in https://matplotlib.org/stable/gallery/animation/double_pendulum.html. The length of the bars is 1 [m], and the positions of the two pendulum weights are observable with Gaussian noise of $\sigma = 0.1, 0.3$, or $0.5$ [m]. The observation interval is $0.03$ [s]. The task is to predict the positions of the two weights in the successive ten frames.

**Network architecture:** $f_\theta$: A sequence of ten "linear blocks" composed of fully connected layers, layer normalizations, and skip connections. Namely, each linear block has three components:

- fc: (input dimension)× (output dimension) linear layer,
- norm: layer normalization,
- skip: skip connection.

Taking four observation variables as input, the first linear block expands the dimensionality to 100. The intermediate linear blocks maintain these 100-dimensional variables. The final linear block reduces the 100-dimensional input to a 50-dimensional output, representing 50 latent space variables. The ReLU activation function is applied throughout the network. The structure of $G_\theta$ mirrors that

Table 4: List of hyperparameters for double pendulum experiment.

| parameter | value |
|---|---|
| $R_{init}$ | diag[1] |
| $Q$ | diag[$e^{-6}$] |
| initial concentration parameter | $e^5$ |

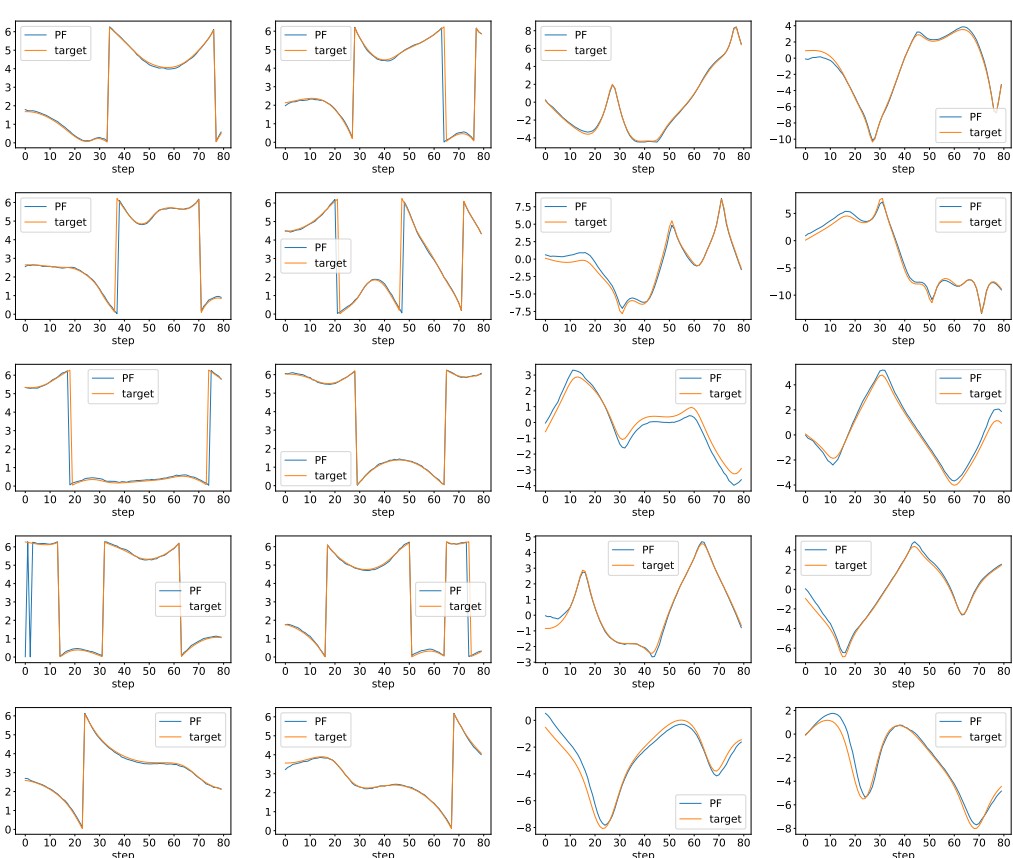

Figure 9: PF results with 100,000 paticles for five example data in test set. Two left columns show evolution of $\theta_1$ and $\theta_2$ (rad) (, therefore, the values are cyclic with the period of $2\pi \simeq 6.3$, and we corrected for those periodic shifts) and the two right columns show $\omega_1$ and $\omega_2$ (rad/s).

of $f_\theta$, while $\phi_\theta$ serves as the inverse of $f_\theta$. The initial eigenvalues are randomly sampled from the range between $e^0$ and $e^{0.01}$.

**Training:** All training variables (network weights for the IOO ($f_\theta$, $G_\theta$), the emission model operator $\phi$, eigenvalues $\lambda$ for the dynamics matrix $A$, Gaussian noise parameter $\sigma$ for angular velocity $\omega$, and the concentration parameter for Von Mises distribution used for angular coordinate $\theta$) are trained together.

**Examples:** Here, we show examples for assimilated $\theta$ and $\omega$ in Fig. 10. Also, we give an additional figure for the RMSE of $\theta$ for various methods.

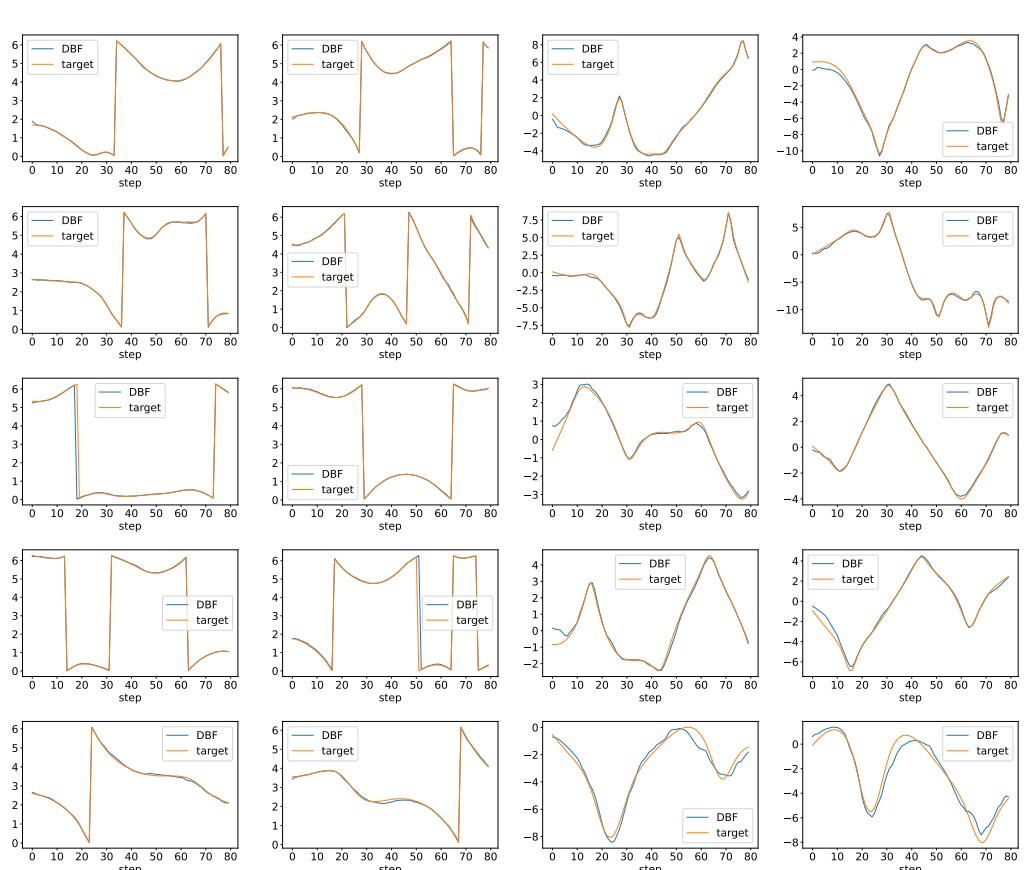

Figure 10: Same as Fig. 9 but for DBF with the latent dimension of 20.

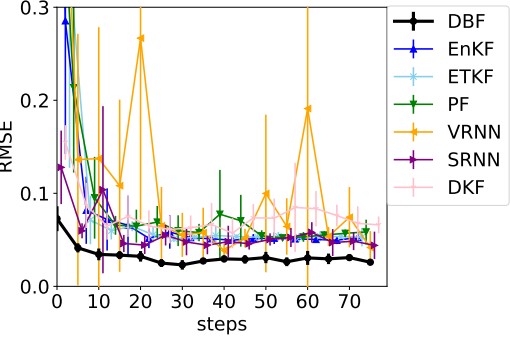

Figure 11: Assimilation results for the angle variable $\theta$. All models successfully determine the angle coordinate in spite of the strong nonlinearity in the observation (trigonometric function). Among these, performance of DBF is the best.

Table 5: List of hyperparameters for Lorenz96 experiment.

| parameter | value |
|-----------|-------|
| $R_{init}$ | diag[1] |
| $Q$ | diag[$e^{-8}$] |

## C.3 LORENZ96

**Dataset:** The dataset consists of physical and observed variables sampled at 40 grid points. The training set includes 25,600,000 initial conditions, while the test set contains 10 initial conditions. The number of training samples is sufficiently large to ensure that the training converges in most cases. The original datasets comprise 80 time steps. Numerical integration is performed using the `solve_ivp` function in SciPy, with a relative tolerance $\text{rtol} = 10^{-2}$ and an absolute tolerance of $\text{atol} = 10^{-2}$. Gaussian noise with standard deviations of $\sigma = 1, 3$, or 5 is added to all measurements.

For KalmanNet, we attempted to train with 25,600,000 and 400,000 initial conditions; however, the process was terminated due to memory limitations. Consequently, we report results using a dataset size of 120,000. For DKF, VRNN, and SRNN, we also tried training with 25,600,000 conditions, but all models encountered a RuntimeError due to instability during the backward computation. To obtain results, we reduced the number of training samples to 512,000. With this adjustment, both SRNN and VRNN successfully completed the training procedure for some initial conditions.

A physical quantity $z_j$ is defined at each grid point $j(1 \leq j \leq 40)$. The time evolution of this quantity is described by the following set of differential equations:

$$\frac{dz(t)_j}{dt} = (z_{j+1} - z_{j-2})z_{j-1} - z_j + F, (1 \leq j \leq 40) \tag{20}$$

In this equation, the driving term $F$ is set to 8. The first term models the advection of the physical quantity, while the second term represents its diffusion along a fixed latitude. With these parameters, the evolution of the physical quantity exhibits chaotic behavior.

**Network architecture:** The NN $f_\theta$ consists of ten convolutional blocks followed by a fully connected layer. Each convolutional block comprises a 1D convolution, layer normalization, and a skip connection:

- conv1d: nn.Conv1d( $c_{\text{in}}$, $c_{\text{out}}$, kernel_size=5, padding=2, padding_mode="circular", )
- norm: layer normalization,
- skip: skip connection.

The first convolutional block has $c_{\text{in}} = 1$ and $c_{\text{out}} = 20$, expanding the input by a factor of 20 in the channel dimension. The subsequent eight layers maintain 20 channels. Finally, the 20 channels and 40 physical dimensions are flattened into 800-dimensional variables, which are then fed into a fully connected layer of size $800 \times 800$. For all layers, the activation function used is ReLU. The function $G_\theta$ is structured identically to $f_\theta$, while $\phi_\theta$ represents the inverse of $f_\theta$.

**Training:** All training variables, including the network weights for the inverse observation operator $f_\theta$ and $G_\theta$, the emission model operator $\phi$, the eigenvalues $\lambda$ for the dynamics matrix $A$, and the Gaussian noise parameter $\sigma$, are trained concurrently.

**Examples:** We show an example figure for assimilation experiment with DBF in Fig. 12.

## C.4 MOVING MNIST (ADDITIONAL LINEAR EXPERIMENT)

**Dataset:** The dataset consists of a series of 2D images, where each pixel has a dynamic range from 0 to 255. The training set contains 480,000 initial conditions, while the test set consists of ten initial

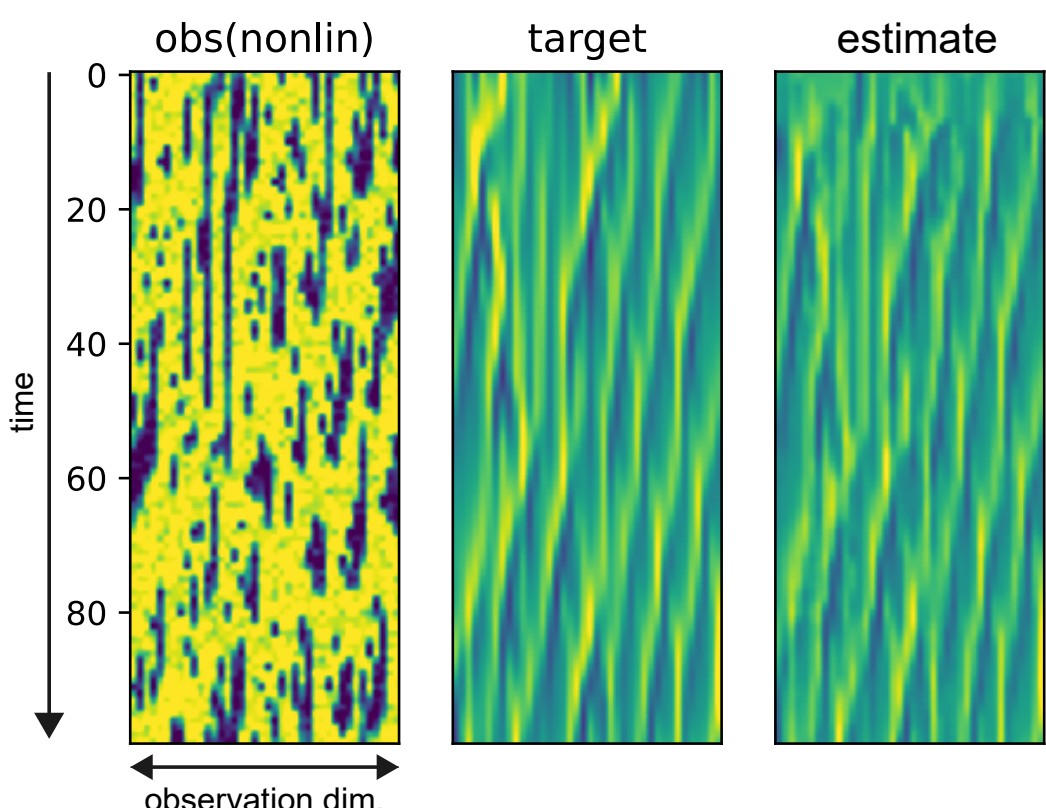

Figure 12: An example of assimilation output in the experiment with nonlinear observation operator. The observation is not very informative due to low threshold for saturation in the observation operator ($o_{t,j} = min(z_{t,j}^4, 10) + \epsilon$,, all cells with $z_{t,j} > 1.8$ are just observed as $10 + \epsilon$). In the first 20 steps, the model output resembles little with the target. However, as the step proceeds, the estimated state begins to capture features of the true state. Even with such a poor observation operator, DBF finds a latent space representation that captures the evolution of the true state.

conditions, with both datasets comprising 20 time steps each. The number of training samples and epochs is sufficiently large to ensure that the training converges effectively. A Gaussian noise with a standard deviation of $\sigma = 50$ is added to all pixels. The MNIST images of the digits "9" (data point 5740) and "5" (data point 5742) move at constant speeds until they reach the edges, where reflection occurs.

**Training:** The network weights for $G_\theta$ are fixed during the first epoch to facilitate the learning of $f_\theta$ and the image tensor for the observation model. Subsequently, $G_\theta$ is trained during the second epoch. In total, DBF undergoes training for two epochs.

**Dynamics model:** Constant velocity model. The exact dynamics matrix we have used is:

$$z_{t+1} = F z_t \tag{21}$$

$$F = \begin{pmatrix} 1 & 0 & 0 & 0 & dt & 0 & 0 & 0 \\ 0 & 1 & 0 & 0 & 0 & dt & 0 & 0 \\ 0 & 0 & 1 & 0 & 0 & 0 & dt & 0 \\ 0 & 0 & 0 & 1 & 0 & 0 & 0 & dt \\ 0 & 0 & 0 & 0 & 1 & 0 & 0 & 0 \\ 0 & 0 & 0 & 0 & 0 & 1 & 0 & 0 \\ 0 & 0 & 0 & 0 & 0 & 0 & 1 & 0 \\ 0 & 0 & 0 & 0 & 0 & 0 & 0 & 1 \end{pmatrix}, z_t = \begin{pmatrix} x_{1,t} \\ y_{1,t} \\ x_{2,t} \\ y_{2,t} \\ v_{x_{1,t}} \\ v_{y_{1,t}} \\ v_{x_{2,t}} \\ v_{y_{2,t}} \end{pmatrix}, \tag{22}$$

and true observation model:

$$\tilde{x}_t = \begin{cases} (x_t \bmod 16) & \text{if x//16 is even} \\ 9 - (x_t \bmod 16) & \text{if x//16 is odd} \end{cases} \text{, same for } y \tag{23}$$

$$o_t = h(z_t), \dim(o_t) = 44 \times 44 \text{ , a } 28 \times 28 \text{ image is embedded at} (\tilde{x}_t, \tilde{y}_t). \tag{24}$$

The formulation above addresses image reflection through the observation operator, resulting in linear dynamics while permitting multiple solutions for each observed figure. This approach presents significant challenges for the EnKF, which assumes a single-peak Gaussian distribution in the assimilating space. To ensure a fair comparison, we revise the dynamics and observation models to allow for a single solution for each figure. This adjustment notably enhances the performance of the EnKF if the image is provided. However, even with this modification, the EnKF fails to accurately estimate the position, velocity, and the embedded image.

**Network architecture:** $f_\theta$: Two-dimension convolutional NNs. Below is the list of layers.

- conv1: nn.Conv2d(1, 2, kernel_size=3, stride=2, padding=1)
- conv2: nn.Conv2d(2, 4, kernel_size=3, stride=2, padding=1)
- conv3: nn.Conv2d(4, 4, kernel_size=3, stride=1, padding=1)
- conv4: nn.Conv2d(4, 4, kernel_size=3, stride=1, padding=1)
- fc: nn.Linear($11 \times 11 \times 4$, 8)

The input image, sized $44 \times 44$, is sequentially processed by convolutional layers (conv1, conv2, conv3, and conv4). The output is then flattened to serve as the input for the fully connected layer (fc). Ultimately, this process yields eight variables for $f_\theta(o_t)$. The network $G_\theta$ follows the same architecture as $f_\theta$, but it produces only the diagonal components of $G_\theta(o_t)$ through the NN.

**Example figures:** In Fig. 14, we show example images for observations and all the algorithms in image-informed setting.

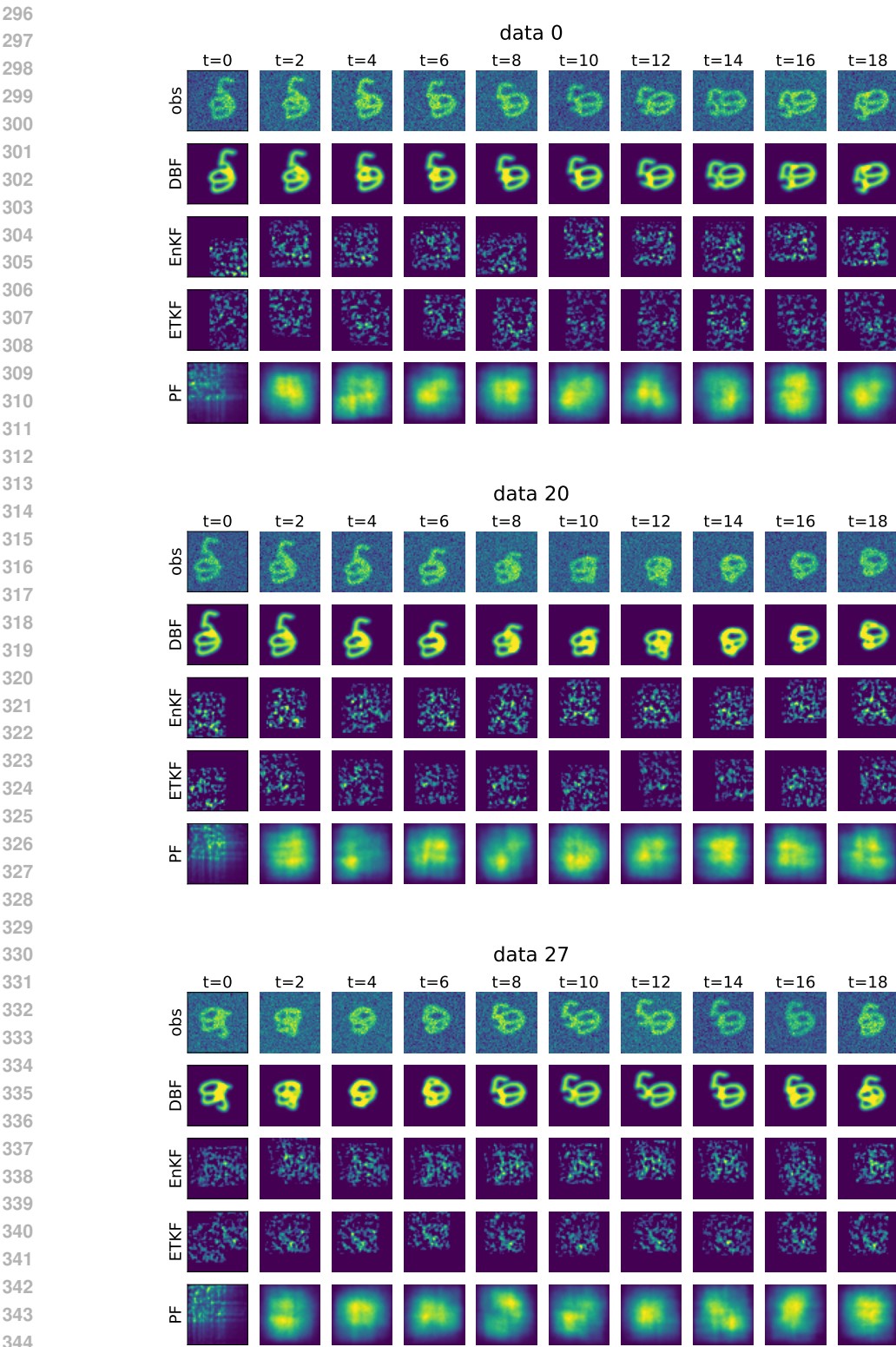

Figure 13: Example figures for two-body moving MNIST experiment. This is the setting explained in the main text. For all algorithms, the two embedded images are not explicitly informed: algorithms need to deal with many unknown parameters in the observation model.

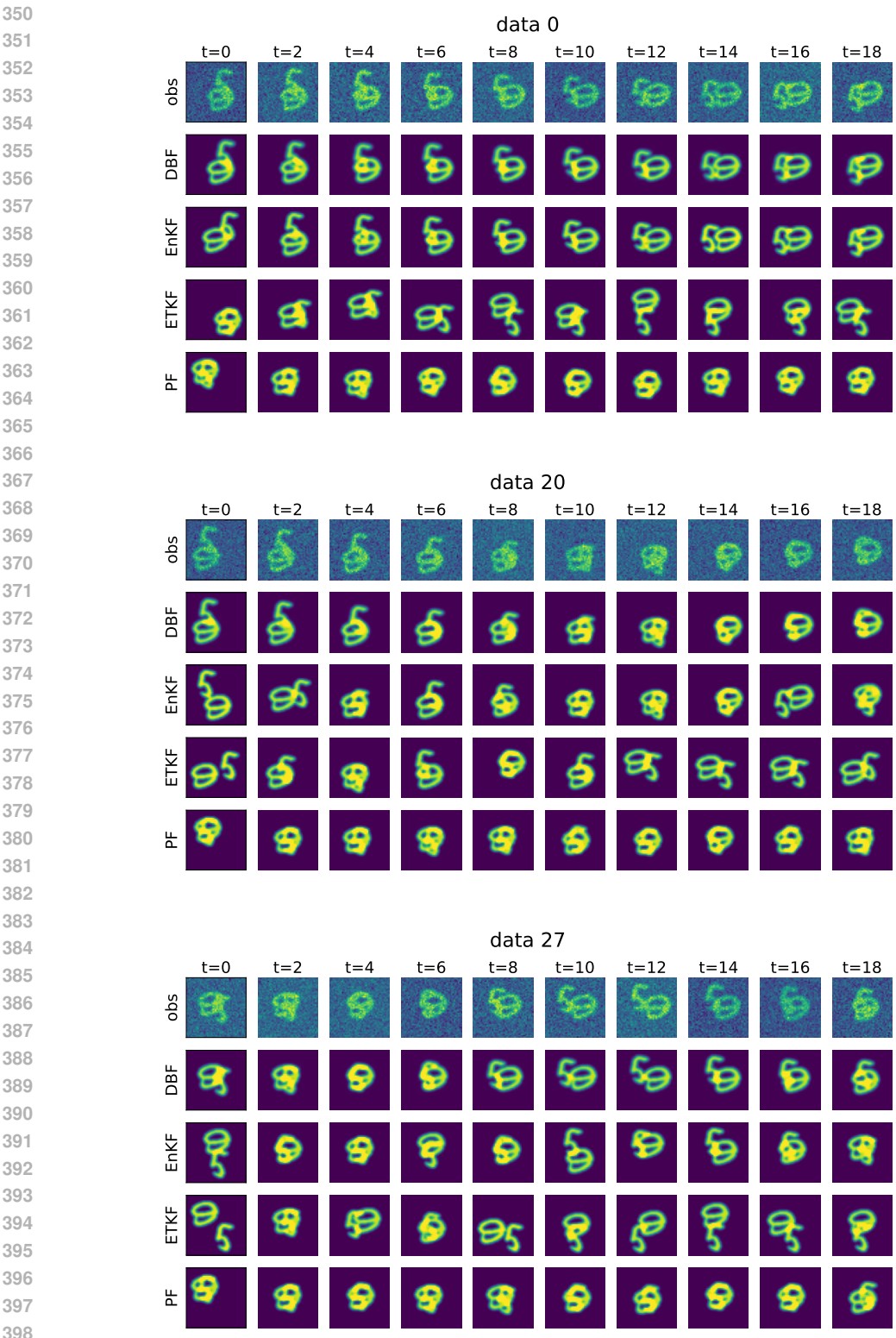

Figure 14: Example figures for two-body moving MNIST experiment. For model-based approaches (EnKF, ETKF, PF), contrary to the experiment reported in the main text, the true images are informed. In data 0, both DBF and EnKF successfully determine and follow the position of the two images. On the other hand, in data 20 and 27, EnKF estimate becomes unstable soon after the two letters overlap. Even in that situation, DBF stably follows the positions of the embedded images.

Table 6: List of hyperparameters for moving MNIST experiment.

| parameter | value |
|---|---|
| $R$ | $\text{diag}[e^6]$ |
| $Q$ | $\text{diag}[e^{-4}]$ |

Table 7: The success rates of different methodologies in the two-body moving MNIST problem. For the model-based approaches, we used the same dynamics and observation models that generated the data. For DBF, the model was initialized with random image tensors and trained solely on the data.

| Method | Success rate |
|---|---|
| DBF | 100% (50/50) |
| EnKF | 58% (29/50) |
| ETKF | 0% (0/50) |
| PF | 0% (0/50) |

## D  TRAINING STABILITY

We observe that the training of our proposed method is stable compared to RNN-based models. Fig. 15 shows the evolution of the real parts of eigenvalues. Although we do not impose constraints on the real parts of eigenvalues, the values only marginally exceed one. Therefore, long-time dynamics is stable during training.

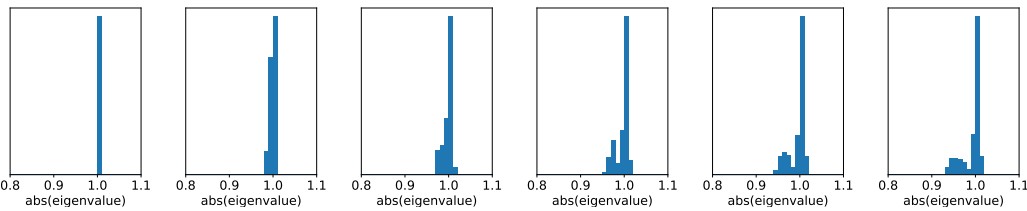

Figure 15: Evolution of histograms for the real parts of 800 complex eigenvalues in Lorenz96 experiment. Initially, eigenvalues are taken as one. As the model learns the dynamics, eigenvalues lower than 1.0 appear. However, the largest eigenvalue $\lambda_{max}$ mostly remains less than 1.02.

## E  HYPERPARAMETER STUDY ON THE LATENT DIMENSIONS

The dimension of the latent variables is a hyperparameter. We have tested the performance and computation (both training and inference) time for nonlinear problems.

### E.1  DOUBLE PENDULUM

#### E.1.1  ACCURACY-COMPUTE TRADE-OFF IN DBF

For double pendulum problem, we test with the standard observation operator with the observation noise of $\sigma = 0.1$. Figs. 16, 17 show the relation between the RMSE and the latent dimensions of the system. Here, we show results with $1.0 \times 10^7$ training data. For the double pendulum problem, we have tested with 4, 20, 80, and 200 latent dimensions. All the latent dimensions tested were too small to observe the impact of the compute-latent dimension relationship. To observe the slowdown, we need to test with higher dimensions. Please also refer to the results for Lorenz96. The performance (RMSE at the final 10 steps) for the angles $\theta$ and angle velocities $\omega$ are poor if the latent dimension

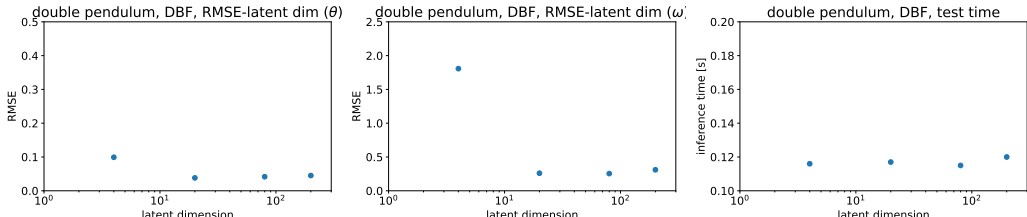

Figure 16: Left panel: RMSE as a function of the latent dimensionality of DBF. Right panel: the inference time as a function of the latent dimensionality of DBF.

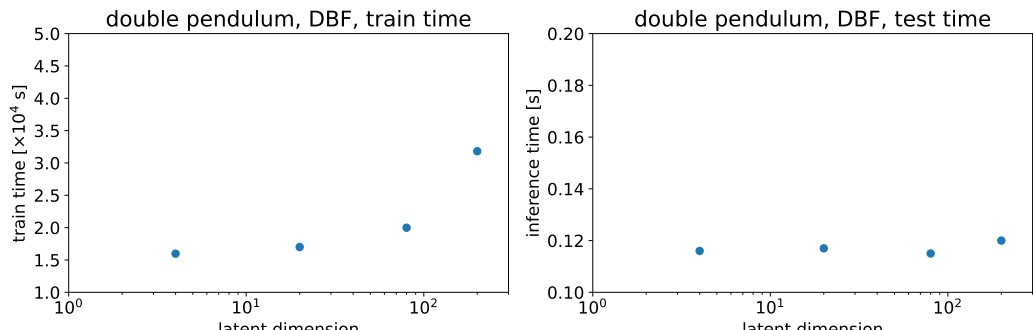

Figure 17: Left panel: the training time for $1.0 \times 10^7$ initial conditions as a function of the latent dimension. Right panel: RMSE as a function of the training time for five different numbers of latent dimensions.

is four. By leveraging 20 latent dimensions, DBF achieves a very good assimilation performance. Further enhancing the latent dimensions to 80 and 200 did not improve performance. The training gradually gets slower when we use latent dimensions higher than 80. As can be seen from Fig. 16, The performance is rather insensitive to the latent dimensions in the range of [20, 200]: the RMSE for $\theta$ is 0.036 at $dim(h_t) = 20$, 0.053 at $dim(h_t) = 80$, and 0.044 at $dim(h_t) = 200$ and for $\omega$ is 0.265 at $dim(h_t) = 20$, 0.375 at $dim(h_t) = 80$, and 0.302 at $dim(h_t) = 200$.

### E.1.2    COMPARISON TO THE PF

The performance of PF depends on the number of particles used. We have tested with 20, 200, 2,000, 20,000, and 100,000 particles. The performance for $\theta$ improves significantly if we use more than 200 particles. The RMSE for the angle velocities $\omega$ almost saturates at RMSE $\simeq 0.31$ when we use particles more than 20,000. To achieve that accuracy, the inference time required for PF is more than 200 seconds per initial condition. On the other hand, DBF achieves slightly better performance (RMSE $\simeq 0.265$) with the latent dimensions of 20. The inference time for DBF is 0.1 seconds per batch.

### E.2    LORENZ96

### E.2.1    ACCURACY-COMPUTE TRADE-OFF IN DBF

For Lorenz96 problem, we test with the nonlinear observation operator with the observation noise of $\sigma = 1$. Figs. 19, 20 show the relation between the RMSE and the latent dimensions of the system. Here, we show results with $1.0 \times 10^7$ training data. The dimensionality of the latent variables can be either larger or smaller than that of the physical variables, but there is a trade-off: up to a certain latent dimensionality, increasing the dimension improves performance at the

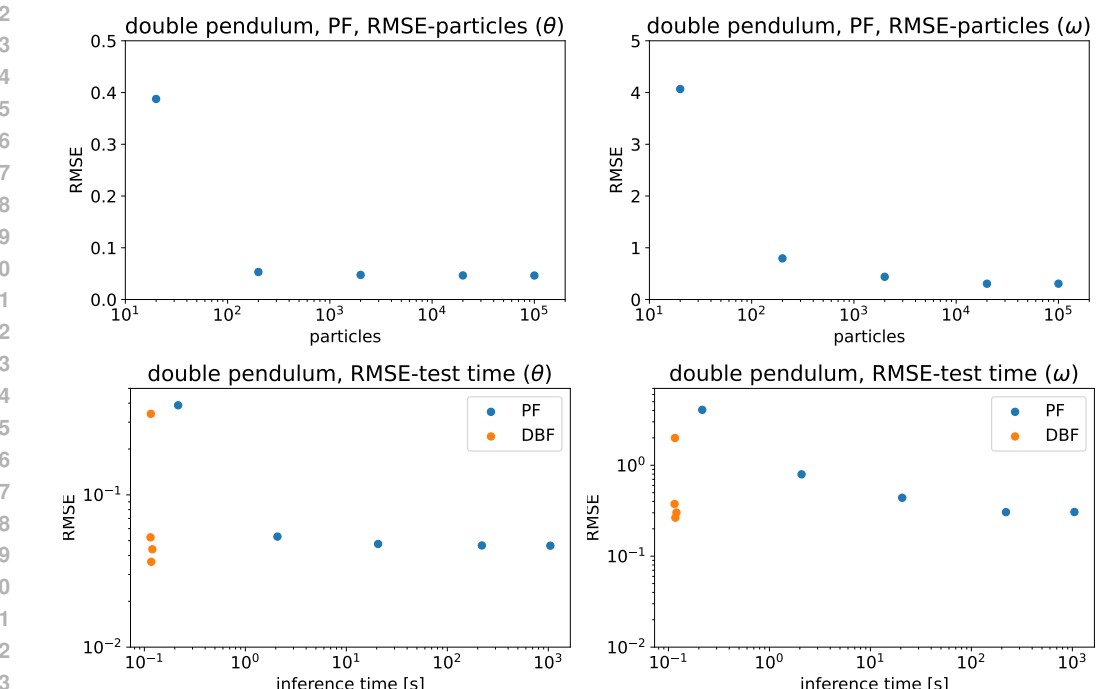

Figure 18: Left panel: the performance of PF as a function of the particles used. Right panel: RMSE as a function of the inference time for the DBF and the PF. For the DBF, the latent dimensions are 20, 80, 200, 800, and 2,000. For the PF, the number of particles are 20, 200, 2,000, 20,000, 100,000.

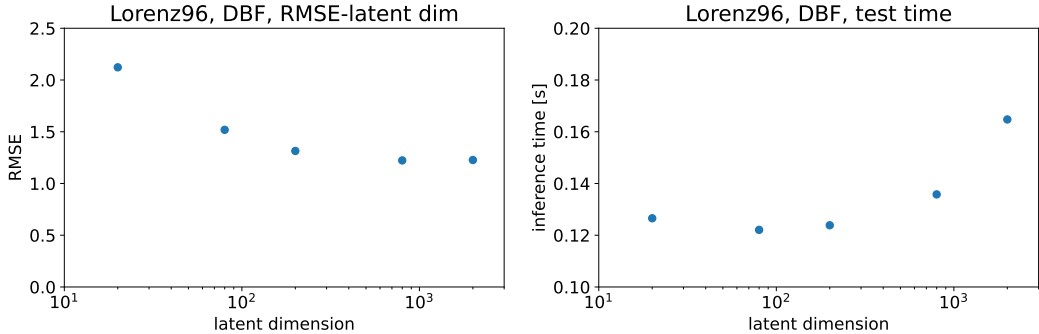

Figure 19: Left panel: RMSE as a function of the latent dimensionality of DBF. Right panel: the inference time as a function of the latent dimensionality of DBF.

cost of longer computation time. Beyond that point, increasing the latent dimensionality no longer improves performance but only increases training time (although inference time remains relatively short compared to model-based approaches). Therefore, the optimal balance depends on the specific problem. For the Lorenz96 system, a dimensionality of 800 was a reasonable trade-off among 20, 80, 200, 800, and 2,000 dimensions. As shown in the figure, the RMSE changes by only 7 percent (1.31 vs 1.23) in the range from 200 to 2,000 dimensions, indicating that the impact is not critical in this range.

### E.2.2 COMPARISON TO THE PF

The PF also has the trade-off. Although RMSE improves slowly as we increase the number of particles, the RMSE was poor (2.27) compared to the DBF results (RMSE $\simeq$ 1.3) even with mas-

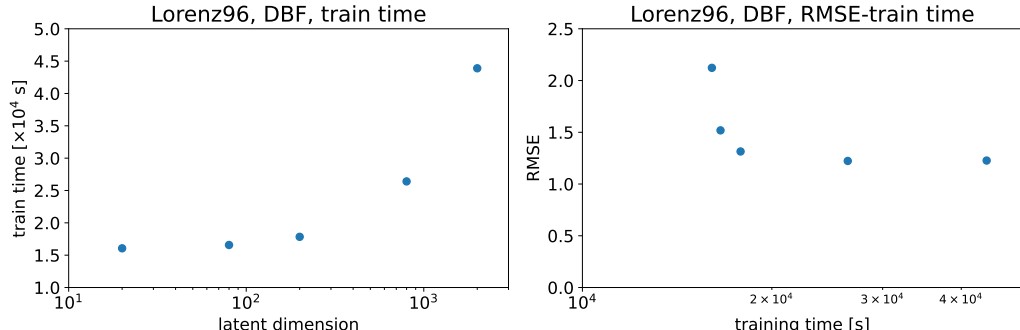

Figure 20: Left panel: the training time for $1.0 \times 10^7$ initial conditions as a function of the latent dimension. Right panel: RMSE as a function of the training time for five different numbers of latent dimensions.

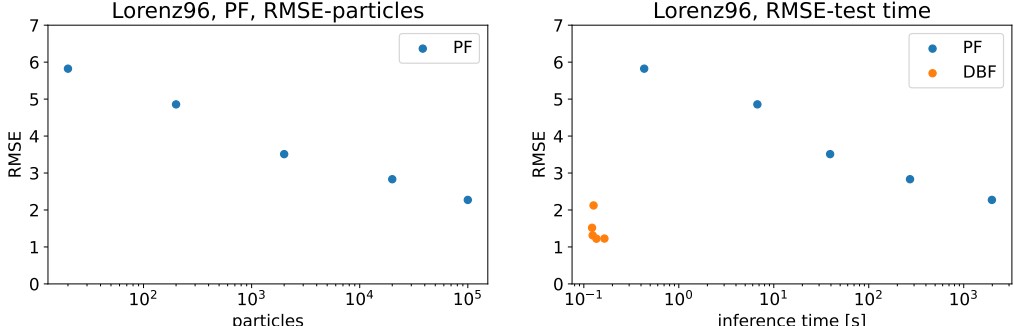

Figure 21: Left panel: the performance of PF as a function of the particles used. Right panel: RMSE as a function of the inference time for the DBF and the PF. For the DBF, the latent dimensions are 20, 80, 200, 800, and 2,000. For the PF, the number of particles are 20, 200, 2,000, 20,000, 100,000.

sively large number of particles (100,000) with very long inference time (2,000 seconds per initial condition)

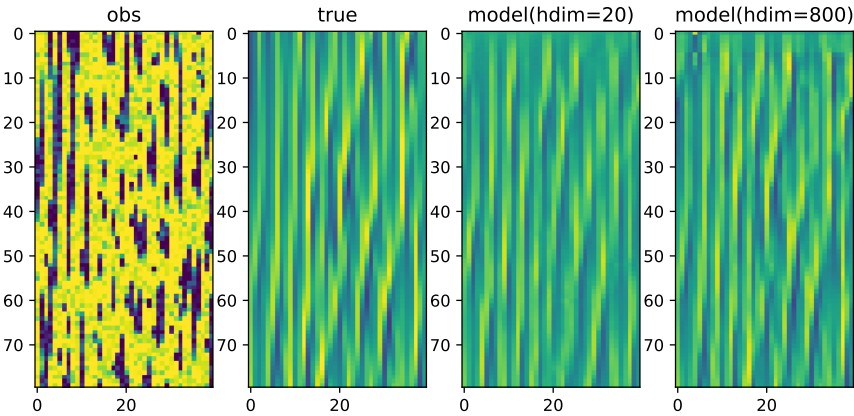

Figure 22: the performance of DBF for a low latent dimension case $(dim(h_t) = 20)$ and a high latent dimension case $(dim(h_t) = 800)$. Even with the latent dimensions (20) smaller than that of the original state space (40), DBF shows the skillful assimilation. With higher latent dimensions (800), the performance further improves.

