# OpenReview forum: "Deep Bayesian Filter for Bayes-Faithful Data Assimilation"
_ICLR.cc/2025/Conference — Submitted to ICLR 2025_

### Official Review · Reviewer_25ic · 2024-11-01

**Soundness:** 2
**Presentation:** 2
**Contribution:** 2
**Rating:** 5
**Confidence:** 4

**Summary:**

The paper presents Deep Bayesian Filtering (DBF), a method for data assimilation within nonlinear state space models (SSMs). It addresses the prevalent challenge of non-Gaussian posterior distributions encountered in many scientific applications. By incorporating new latent variables, DBF ensures Gaussianity in the latent space, facilitating linear transitions and inverse Gaussian observation operators. This approach claims that one of the advantages is to avoid errors associated with successive Monte Carlo sampling, as seen in DVAEs.

**Strengths:**

The paper is well-structured, making it easy to follow the ideas presented. The proposed method can be considered to be novel and could be particularly beneficial for certain generic high-dimensional dynamical systems, especially those that can offer underlying latent states for training.

**Weaknesses:**

The primary limitation, in my view, is that the model necessitates extensive training data, including the underlying system's latent states (nonlinear dynamics), to effectively learn the IOO. This requirement could render DBF less practical for real-time applications or in domains where data availability is limited, making model-based methods potentially more viable. Additional issues and questions are discussed in the Questions part.

**Questions:**

1. A discussion (or even a comparison) between the normalizing Kalman filter (NKF) and also the model mentioned in the DVAE paper, Kalman Variational Autoencoders (KVAEs), seems necessary, as in these two models, the generative structure is the same as the one shown in Panel (b), Figure 1.

     - de Bézenac E, Rangapuram SS, Benidis K, et al. *Normalizing Kalman Filters for Multivariate Time Series Analysis*. *NeurIPS*, 2020;33:2995-3007.

2. The loss functions in Equations 6 and 7 still require Monte Carlo sampling if $p(o_t | h_t) $ is non-Gaussian. While DVAEs must sample the latent state trajectory due to the variational distribution is indeed model the smoothing distribution, the benefit of “Bayes-faithful” data assimilation highlighted here is not entirely clear. If Monte Carlo sampling is still required at each step, and if the neural network is shared and amortized over each time step $t$ (which is the same as in DVAEs), what is the specific advantage of this “Bayes-faithful” property? Are there empirical results demonstrating the practical benefit of this property?

3. Follow the question 2, the paper claims to present the first VAE-based model for time-series data that maintains a posterior structure faithful to the Markov property in SSMs. However, recent work on Auto-EnKF models also employs filtering distributions to construct the model evidence or ELBO. For example, Equation (32) in Lin et al. (2024), shares a similar loss function structure to Equation (6) in this paper. Including a discussion or comparison of these models would enhance the comprehensiveness of the proposed method's positioning.

     - Chen Y, Sanz-Alonso D, Willett R. "Autodifferentiable Ensemble Kalman Filters," *SIAM Journal on Mathematics of Data Science*, 2022; 4(2):801-33.

     - Chen Y, Sanz-Alonso D, Willett R. "Reduced-order Autodifferentiable Ensemble Kalman Filters," *Inverse Problems*, 2023; 39(12):124001.

     - Lin Z, Sun Y, Yin F, Thiéry AH. "Ensemble Kalman Filtering Meets Gaussian Process SSM for Non-Mean-Field and Online Inference," *IEEE Transactions on Signal Processing*, 2024 Aug 22.



4. Other questions:

 - Adding derivations for Equations (1, 2, 4, and 5) in the Appendix would improve clarity.

 - Could the authors discuss potential applications for the proposed method?

 -  In Section 3.1, the authors mention that “DVAEs are not comparable as they need to undergo supervised training,” which may need clarification. If I’m not mistaken, DVAEs typically require only observation data and do not rely on model knowledge or latent state data. Could you clarify why DVAE training is interpreted as supervised in this context?

 - Given the low latent dimension and the known model structure, it seems model-based methods (e.g., EnKF, PF) should perform reasonably well. Could you explain why DBF and other data-driven methods outperform these model-based approaches in Table 1?

---

> ### Author Response · Authors · 2024-11-22
> **Thank you for the review! zt is not needed at inference time, therefore does not inhibit DBF from real-time applications**
>
> Thank you for the constructive comments. We give answers to the comments below.
>
> *“The primary limitation, in my view, is that the model necessitates extensive training data*
> - **See the comments for all reviewers:** in real-time applications, inference time is the most critical factor. A trained DBF is able to give the inference only from observation o within 0.13 seconds (for Lorenz96 problem, 800 latent dimension). The physical variables zt are required only for training and there is no need to use zt when inference.
>
>     Given that we have the state space model, we are able to produce the dataset for training so that we can train DBF with the synthesized dataset. This problem setting, where data assimilation is performed using a known state-space model, is typical in natural sciences such as meteorology and oceanography. We believe that the contribution of our methodology is significant as a methodological paper.
>
> *A discussion (or even a comparison)...*
> - Thank you very much for the reference. Probabilistic models of NKF and KVAE are indeed similar to ours. While it shares the goal of learning state-space models with linear dynamics and nonlinear observations, the methods are not aimed at data assimilation (i.e., their model does not try to infer the physical variable zt but only to reproduce distribution of observations). We will discuss this in the revised manuscript.
>
> *The loss functions in Equations 6 and 7...*
> - The crucial difference is that the MC samplings in DBF are not nested with each other. In DVAE, we need to evaluate an integral term $\int q(h_{1:T}|o_{1:T}) \log p(o_{1:T},z_{1:T}) do_{1:T},h_{1:T}$, where $q(h_{1:T}|o_{1:T})＝\prod_t q(h_t| h_{t-1}, o_t )$. Although the log-term could be factorized as $\sum_t (\log p(o_t | h_t) + \log p(h_t | h_{t-1}))$ thanks to the Markov property, we need MC (nested) sequential sampling over $h_{1:T}$ if we want to evaluate the term at $t=T$. On the other hand, ELBO in DBF is $\sum_t \int q(h_t|o_{1:t}) \log p(z_t|h_t) dh_t + KL[q(h_t|o_{1:t})|q(h_t|o_{1:t-1})]$ because DBF takes the lower limit of $\sum_t \log p(o_t|o_{1:t-1})$. Thanks to the analytic expressions of $q(h_t|o_{1:t})$ and $q(h_t|o_{1:t-1})$, the KL term can be computed analytically. A MC sampling is needed to compute $\int q(h_t|o_{1:t}) \log p(z_t|h_t) dh_t$ but this is independent from other timesteps. We will add this detailed explanation in the appendix of the revised manuscript.
>
> *Follow the question 2,...*
> - In our paper, we call DBF faithful to Markov because it computes the filtering distribution by recursive formula derived from the Markov property. Although Auto-EnKF and other models such as Lin+24 indeed uses the evidence lower bound as the loss function, the procedure for the construction of the posterior is different from our method. We will clarify this point and rewrite the main text accordingly.
>
> *Adding derivations for Equations ...*
> - Thank you for pointing out. We will add derivations in the appendix.
>
> *Could the authors discuss potential applications ...*
> - For example, in numerical weather forecasting, we have a numerical simulation code. With the code and observation operator that is being used for the operational weather forecasting system, we prepare the pairs of (zt, ot) data. Using the dataset, we train DBF. With the trained DBF and real observation data obtained from, for instance, infrared satellites. The resultant assimilation system would be highly efficient because of large non-Gaussianity in the true posterior in physical space.
>
> *In Section 3.1,...*
> - DVAE generates latent variables, but the variables are different from the state variables of the original state-space model provided in the problem. Therefore, while DVAE can predict the time-evolved images themselves, it cannot infer the position or velocity from those images. This is what we referred to as 'not comparable.' While it is possible to compare the accuracy of image prediction itself, we expect that the long-term predictions would be poor as DVAEs are not based on known dynamics.
>
> *Given the low latent dimension...*
> - The low latent dimension does not (necessarily mean guarantee) that model-based methods work with sufficient accuracy. In EnKF, assumption of Gaussianity over the physical space would induce substantial bias, particularly in highly chaotic problem setting as double pendulum. Although PF should not induce such bias (and indeed, it performs well for $\theta$ (the positions) in low-noise settings), the results for $\omega$ (the velocities, not directly observed) were still poor compared to DBF even with an experiment with 20,000 particles (see **PF_particles_doublependulum_{omega/theta}.png in the supplementary material**). We suspect that the number of particles remains insufficient for a 4-dimensional latent space:
> $20000^{1/4} \simeq 12$ particles are inadequate to effectively sample the physical space, especially given the highly chaotic dynamics of the double pendulum.

---

> > ### Comment · Reviewer_25ic · 2024-11-22
> >
> > Thank you to the authors for their response. After reviewing the feedback, I find myself in agreement with most points.
> >
> > However, if you emphasize the necessity of underlying state data, it may be beneficial to discuss the KalmanNet and EKFnet series, as these works also utilize data to develop efficient and accurate filtering algorithms. Given the abundance of related research in this field, it would be advantageous for the authors to clearly outline the problem setup and engage with existing literature in the revised manuscript to effectively highlight this paper's unique contribution. Additionally, I suggest including a limitations/future work section to address any limitations of the proposed method, if necessary.
> >
> > - Revach G, Shlezinger N, Ni X, Escoriza AL, Van Sloun RJ, Eldar YC. KalmanNet: Neural network aided Kalman filtering for partially known dynamics. IEEE Transactions on Signal Processing. 2022 Mar 11;70:1532-47.
> >
> > - Xu L, Niu R. EKFNet: Learning System Noise Covariance Parameters for Nonlinear Tracking. IEEE Transactions on Signal Processing. 2024 Jun 20.

---

> > > ### Author Response · Authors · 2024-11-25
> > >
> > > Thank you for your prompt response. We would like to respectfully clarify two points from our original text:
> > > 1. KalmanNet is already discussed and included in our performance comparison for nonlinear problems, and
> > > 2. a limitations section is already provided.
> > >
> > > We have outlined the problem in detail in Section 2.1: Inference of physical variables in a state-space model and discussed the positioning of our work in Section 2.6: Related works.
> > >
> > > We sincerely appreciate the reference to EKFNet and will include it in the revised manuscript.
> > >
> > > If you find our approach promising and feel that your concerns have been addressed, we kindly request you to consider raising the score.

---

> > > > ### Comment · Reviewer_25ic · 2024-11-28
> > > >
> > > > Thanks to the authors for the update on the manuscript. But, it is also important to clarify the arguments similar to the following:
> > > >
> > > >        DBF is the *first* VAE-based model for time-series data that maintains a posterior structure faithful to the Markov property in SSMs.
> > > >
> > > >       DBF is the first “Bayes-Faithful” amortized variational inference methodology, constructing test distributions that mirror the inference structure of a SSM with the Markov property.
> > > >
> > > > As previously mentioned, the loss and the proposed Bayesian-faithful methods are not the "first" ones.

---

> > > > > ### Author Response · Authors · 2024-11-29
> > > > >
> > > > > Thank you for the comment. While we initially believed that the phrase
> > > > >
> > > > > *'DBF is the first VAE-based model for time-series data that maintains a posterior structure faithful to the Markov property in SSMs.'*
> > > > >
> > > > > effectively conveyed the intended meaning, we appreciate the feedback and will revise to
> > > > >
> > > > > *`DBF is the first ``Bayes-Faithful'' amortized variational inference methodology, constructing posteriors that replicate the inference structure of a SSM with the Markov property.'*
> > > > >
> > > > > in the camera-ready version to improve clarity.

---

> ### Comment · Reviewer_25ic · 2024-11-26
>
> Thank you for the clarification and for addressing my concerns.
>
> I appreciate the detailed explanation regarding KalmanNet and the limitations section. I apologize for overlooking these points in my last comment. After reviewing the updated paper and the public discussions, I would like to raise my score to a 6 to reflect my current view of this paper.

---

> ### Comment · Reviewer_25ic · 2024-11-29
>
> I still cannot agree that this work is the "first" to use the amortized VI method to construct a Markovian posterior for latent states. For me, a very similar structure is the EnVI from Lin et al. (2024), as mentioned, which used filtering distribution to approximate the variational distribution over latent states, and the model parameters are learned through amortized learning.
>
> In addition, the updated manuscript contains an assertion that "Auto-EnKF (Chen et al., 2022; 2023) leverages EnKF and trains the model by optimizing ELBO." This statement is absolutely incorrect, as Auto-EnKF actually trains the model by maximizing the model evidence, not the ELBO, i.e., there is no VI approximation for the latent states. But these works are very competitive in terms of model learning and latent state estimation.
>
> It appears that the authors have not fully addressed my initial comments, nor have they thoroughly reviewed the relevant literature.

---

> > ### Author Response · Authors · 2024-11-29
> >
> > Thank you for your comment.
> >
> > We would like to clarify the contributions of our proposed method. Our central claim is that DBF is a posterior distribution estimation method leveraging the Markov property of the original state-space model under the following conditions: (i) the dynamics function of the original state-space model is deterministic, and (ii) the Koopman operator can embed these dynamics into linear dynamics.
> >
> > DBF addresses the challenge of estimating the posterior distribution over a latent variable sequence within a given state-space model. Existing DVAE approaches typically rely on RNNs to compute posteriors and optimize an ELBO expressed as $\int q(h_{1:T}|o_{1:T}) \log p(o_{1:T},z_{1:T}) do_{1:T},h_{1:T}$. However, these methods represent the posterior only in sample form, leading to cumulative approximation errors due to Monte Carlo sampling. In contrast, our approach optimizes an ELBO expressed as $\sum_t \int q(h_t|o_{1:t}) \log p(z_t|h_t) dh_t + KL[q(h_t|o_{1:t})|q(h_t|o_{1:t-1})]$ By constraining the latent space dynamics to be linear and performing update steps via a learnable IOO, our method avoids the accumulation of approximation errors. Even when the original dynamics are nonlinear, they can be represented as linear dynamics in a sufficiently high-dimensional latent space using the Koopman operator. Thus, the linearity assumption is not a restrictive limitation. Under the embedded dynamics, our approach prevents the accumulation of sampling errors.
> >
> > To further clarify, we compare our method with Lin+24, which addresses a fundamentally different problem setting. Lin+24 models dynamics using a Gaussian Process (GP). While the original state-space model involves variables f, x, and y, Lin+24 approximates the GP with reduced computational cost by introducing representative points and solving for a posterior distribution over u instead of f. This process results in solving for a posterior distribution associated with a problem distinct from the original state-space model. Additionally, Lin+24 employs approximations such as $p(x_{t−1}|u,y_{1:t−1}) \simeq p(x_{t−1}|u,y_{1:t})$ and $q(x_t|u,x_{t−1}) \simeq p(x_t|u,x_{t−1},y_{1:t})$, which our methodology does not require. For these reasons, Lin+24 is not designed to be a VAE that respects the Markov property of the original state-space model.
> >
> > Regarding the Auto-EnKF, it optimizes an approximation of the log model evidence, not the ELBO. Thank you for pointing this out; we will address this in the camera-ready version.

---

> > > ### Comment · Reviewer_25ic · 2024-12-02
> > >
> > > Thanks to the authors for their reply. Let me refocus our discussion on the core issue: whether the posterior distribution design proposed in your paper is truly **the first** to adhere to the Markov property of the original SSM. I believe the paper overstates this claim, as this concept is not entirely novel. My evidence is that, for instance, Lin et al. (2024) also propose a variational distribution that respects the Markov property of the original SSM.
> > >
> > > Upon revisiting Lin et al. (2024) and your paper, I’d like to reclarify my position:
> > >
> > > - While Lin et al. (2024) and your work address slightly different problems, as you mentioned, Lin et al. utilize Gaussian processes (GPs) to model system dynamics, requiring additional posterior inference. However, the discussion focus is on the design of the variational distribution. Below, I copy their loss formulation shown in Eq. (32) their paper:
> > > $$ L_{\ell_t}= E_{q(u)}\left[ E_{q(x_t \vert u, y_{1:t})} (\log p(y_t \vert x_t))\right] - E_{q(u)}\left[ KL(q(x_t \vert u, y_{1:t}) \| p(x_t \vert u, y_{1:t-1}) )  \right] - KL(q(u) \| p(u))
> > > $$
> > > where $x_t$ denotes latent state, $y_t$ is observation, $u$ denotes GP-based system dynamics surrogage, and the variational distribution $q(x_t \vert u, y_{1:t}) \approx p(x_t \vert u, y_{1:t}) $ is constructed using an ensemble Kalman filter, avoiding the need for the sampling as in DVAEs and definitely adhering strictly to the Markov property of the SSM. Once the Gaussian process for system dynamics is learned (i.e. we have $q(u)$), is this formulation comparable to Equation (6) in your paper, which employs IOO to construct the posterior distribution instead?
> > > Thus, how can it be asserted that your work is **the first** to design a posterior distribution respecting the Markov property of the original SSM?
> > >
> > >
> > > **Response 1:**
> > >
> > >      We would like to clarify the contributions of our proposed method. Our central claim is that DBF is a posterior distribution estimation method leveraging the Markov property of the original state-space model under the following conditions: (i) the dynamics function of the original state-space model is deterministic, and (ii) the Koopman operator can embed these dynamics into linear dynamics.
> > >
> > > - This claim is reasonable, as it no longer asserts it is "**the first**" work to leverage the Markovian property to design the variational distribution.
> > > - While you emphasize the use of the Koopman operator, its practical implementation remains theoretical. For instance:
> > >
> > >   - The dimensionality of $h_t$ and the learned basis functions for the Koopman operator via neural networks are uncontrollable. This also partially addresses reviewers’ concerns regarding the identifiability of $z_t$. My guess is that, in this work, introducing two neural network-based approximations means the identifiability cannot be guaranteed or characterized at this moment.
> > >
> > >
> > > **Response 2:**
> > >
> > >     DBF addresses the challenge of estimating the posterior distribution over a latent variable sequence within a given state-space model. Existing DVAE approaches xxxxxxxxxxxxxxxxxxxxxxxxxxxxxx xxxxxxxx xxxxxxxxxThus, the linearity assumption is not a restrictive limitation. Under the embedded dynamics, our approach prevents the accumulation of sampling errors.
> > >
> > > - I am fine with this claim but maintain minor concerns about the use of Koopman operator as mentioned previously.
> > >
> > > **Response 3:**
> > >
> > >     xxxxxxx. This process results in solving for a posterior distribution associated with a problem distinct from the original state-space model.
> > >
> > > - This depends largely on the information contained in the observation data and the observation function of the original SSM. Some examples in my mind are:
> > >
> > >    - If the observation dimension is higher than that of the latent state, it should be generally sufficient, though non-identifiability may occur.
> > >
> > >    - If the observation function $g(\cdot)$ is invertible and known, the GP-learned system dynamics should effectively capture the underlying dynamics.
> > >
> > >    - If the underlying latent states are accessible (in the same setting as in this paper), the GP is fully capable of representing the system’s dynamics.
> > >
> > > **Response 4:**
> > >
> > >     Additionally, Lin+24 employs approximations such as p(xt−1|u,y1:t−1)≃p(xt−1|u,y1:t) and q(xt|u,xt−1)≃p(xt|u,xt−1,y1:t), which our methodology does not require. For these reasons, Lin+24 is not designed to be a VAE that respects the Markov property of the original state-space model.
> > >
> > > -  As I mentioned earlier, these assumptions ultimately lead to approximating the variational distribution $q(x_t \vert u, y_{1:t})$ using the filtering distribution derived from the EnKF, which adheres to the Markov property of the original SSM.

---

> > > > ### Author Response · Authors · 2024-12-02
> > > >
> > > > Thank you for your response.
> > > >
> > > > Our statement, “DBF is the first Bayes-Faithful VAE,” was intended to convey that it is the first DVAE method to construct an approximation of the posterior distribution through prediction and update steps derived from linear dynamics. As acknowledged by all other reviewers, our method for constructing this posterior distribution is novel. However, as you pointed out, our explanation was misleading. In the camera-ready version, we will revise it to a more accurate description: “DBF offers a novel ‘Bayes-Faithful’ approximation for the posterior within the dynamical VAE, constructing the posterior following the sequential Bayesian update formula.”

---

### Official Review · Reviewer_b7Ma · 2024-11-01

**Soundness:** 2
**Presentation:** 2
**Contribution:** 3
**Rating:** 5
**Confidence:** 3

**Summary:**

The paper proposes a variational inference approach “DBF” to data assimilation, i.e. the task of obtaining the posterior over an SSM state $z_t$ given observations $o_{1:t}$. In contrast to the linear Kalman filter, DBF supports non-linear observation models, as well as non-linear SSMs. In contrast to related dynamical VAEs (DVAEs), DBF does not rely on RNNs and can be trained more easily (less training instabilities).

DBF assumes a latent state $h_t$ that evolves linearly, with a Gaussian variational distribution for the approximate posterior $q(h_t)$, ensuring that the approximate posterior remains Gaussian with analytic updates under new observations. When the true SSM dynamics are linear, $h=z$ and only the (nonlinear) mapping from observation $o_t$ to latent state $h_t$ is learned.

When true SSM dynamics are non-linear, then $h$ is a latent state with higher dimensionality than $z$, but linear dynamics $h_{t+1}=A h_t$; DBF learns these latent dynamics $A$ as well as a latent emission model from the linear state $h_t$ to the actual SSM state $z_t$ from pairs $(z_t, o_t)$ of samples of SSM state $z_t$ and corresponding observation $o_t$.

The form of the variational approximation matches the Markov properties of the underlying SSM, hence termed “Bayes-faithful”.

The paper compares DBF to classical data assimilation algorithms (e.g. ensemble Kalman filter, particle filters) and DVAE variants in three settings (1. linear dynamics on 8-dimensional state, nonlinear observation with 784 dimensions; 2. nonlinear dynamics on 4-dimensional state, nonlinear mixed observation with 2 dimensions; 3. nonlinear dynamics on 40-dimensional state, nonlinear point-wise observation).

Amongst all compared methods, DBF consistently performs best in most settings (the one exception being when the state is directly observed with low noise, thus true posteriors are almost Gaussian and ensemble Kalman methods perform best).

**Strengths:**

Combining a neural network model to learn nonlinearities in observation and state whilst assuming an underlying _latent_ state with linear dynamics seems original (I am not very familiar with the literature). This fills a gap between hard-to-train DVAEs, Kalman filter approaches that struggle with non-Gaussianity, and particle filters that do not scale to high dimensions. If this holds up in larger-scale experiments, it will be a significant contribution to data assimilation in applications such as weather models.

The paper clearly states the approach’s limitation to require pairs of samples and observations.

In the experiments, a wide range of different methods are included as baselines. The supplementary material provides a high level of detail on experiment settings for DBF itself (though not on the baselines).

**Weaknesses:**

The main weakness of this paper is that the empirical evaluation insufficiently supports the claims made about the proposed DBF.

While DBF’s assumed Gaussianity and linearity in latent space leads to analytic solutions, this can be considered to do closed-form inference in an _approximate_ model. As noted in the paper, given a sufficient number of particles, a particle filter would be able to represent the true posterior. However, the paper does not directly analyse the loss in accuracy due to the approximations made in DBF’s variational inference approach. While it does compare to particle filters, the paper only considers a single, fixed number of particles, and simply states that it is not sufficient. The evaluation that would be needed here is to compare the accuracy–compute trade-off. For particle filter, how do accuracy and resource needs scale with number of particles, and correspondingly, for DBF, how do accuracy and resource needs scale with the number of latent dimensions? Ideally, for a range of dimensions of the actual SSM.

While the paper comes with code for the DBF implementation in the supplementary material, the main text itself is short on detail of the method. For example, the ELBO training objectives are simply given, without explicit derivation.

While ‘Strategy 1’ is introduced to train solely with samples of $z_t$, not requiring pairing with observations, this is not actually included in the empirical evaluation.

**Questions:**

1. Is your approach identifiable? I.e., is there a unique latent (linear dynamics) state trajectory corresponding to a given physical trajectory? If not, what effect does the nonidentifiability have?

2. Can DBF represent multimodal posteriors?

3. line 273: “computations for covariance matrices become challenging in high-dimensional spaces” – can you explain why these are not needed in DBF?

4. One of your key motivations seems to be that particle filters don’t scale to high dimensions, and DBF would scale better to high dimensions. But then you also need to have a significantly larger latent dimension than the actual state space to capture the nonlinear state transitions through the linear embedding: does it still reduce computational cost overall?

5. Also, how do the _state_ dimensions vs the _observation_ dimensions affect the performance of DBF vs particle filters?

6. You mention Frerix et al. (2021), where the Inverse Observation Operator is also learned – could you have included their approach in the empirical comparison?

7. How did you set up & run the baselines – can you link to which implementation you used?

8. Baseline methods could not handle the amount of training data used for DBF. To what extent is better performance due to being able to ingest more data, i.e. how would DBF have performed at the same amount of training data as baselines?

9. line 353: “We define success as …” seems rather arbitrary, and this might bias results (surprisingly related: “Are Emergent Abilities of Large Language Models a Mirage?” by Schaeffer et al., NeurIPS 2023), what are the RMSE results themselves?

10. line 365: can you explain how learning system parameters sets DBF apart from traditional approaches? I am not intimately familiar with the field but surely there are other approaches that learn system parameters.

11. I would have liked to see some sample trajectories vs inferred of the different methods.

12. line 480–485: could you quantify the (non)Gaussianity?

13. I would have liked to see an ablation study; for example, how does the choice of latent dimension affect the accuracy and compute requirements of your approach?

## Clarity questions

14. line 90: What do you mean by “regression methods” here? In general, regression can be probabilistic as well, thus quantifying uncertainty and not only providing point estimates.

15. p.3, Eq. (1): $\mu_t$ shows up in the last term on the right-hand side; is this a mistake? What would be the correct equation?

16. line 145: what do you mean by “virtual prior”, can you explain this in more detail?

17. Eq. (6): what is the KL term of the first element in the sum ($t=1$), where there are no previous observations?

18. Would you call DBF part of the DVAE family or not? In line 224 you say “DVAEs are closely related” (implying no), in line 255 you write “apart from other DVAEs” (implying yes).

19. line 265: “LS4 replaces recurrence with convolutions, forgoing a recursive approach” - does this matter?

20. $\sigma$ in line 339 vs $\sigma_{sys}$ (NB- should be $\sigma_\text{sys}$) in line 346: I don’t understand the distinction, can you explain?

21. line 348: How can the input dimensionality exceed the GPU memory – should this not also depend on sequence length, batch size, etc.?
(NB- it was unclear from the main text where the $44 \times 44$ came from, only explicitly mentioned in supplementary. It is especially confusing as in line 359 you now discuss dimensionality of digits themselves, should this not be irrelevant?)

22. Fig. 2 (b): what are the iterations – training steps of NN ?

23. Fig. 3 (b): is this RMSE of $\omega$ alone (averaged over both velocities, see below)?

24. Table 1 and line 420, “RMSE between physical variables”: RMSE is given for $\theta$ and $\omega$ separately, but there are two of each: is this the average over the two angles / velocities?

25. line 422, excluding failed initial conditions: what is the fraction of initial conditions that failed?

26. line 426: should this be time steps (of dynamics, as opposed to training steps of model)?

27. lines 483, 485: what do you mean by “prior $q(z_t | o_{1:t-1})$” vs “prior $p(z_{t+1} | o_{1:t})$”? Are they the same (except for the shift by 1)? Why $p$ vs $q$?

## Further points for improving clarity

You refer to the “test distribution” $q(z_t)$; in variational inference, this would more commonly be referred to as ‘approximate posterior’ or ‘variational distribution’ – mentioning this equivalence explicitly would make the paper a bit more accessible.

line 62, “assuming Gaussianity”: also mention explicitly, already in the introduction, the assumption of linear dynamics for the latent variables $h$.

line 156: “LGSS” is used but not introduced what it stands for.

line 288: citations for Lorenc and Frerix should be in parentheses.

line 334, “bouncing off frame edges”: at first reading I was quite confused how this clearly non-linear behaviour would match up with underlying _linear_ dynamics; it became clear from reading the supplementary, but would be helpful to mention here already that this nonlinearity is part of the _observation_ model, not the state.

Fig. 4: “for one of data” sounds a bit odd

---

> ### Author Response · Authors · 2024-11-22
> **Thank you very much for the detailed review: see the updated supplementary material for accuracy-computation trade-off**
>
> Thank you very much for the constructive review. We are happy that you have acknowledged the promise of this methodology. We give answers to questions and comments below.
>
> *While DBF’s assumed Gaussianity and linearity...*
> - As you have pointed out, the trade-off between accuracy and computation is crucial. **See the updated supplementary material for figures on the trade-off.**
>
>     Given unlimited computational resources, the Particle Filter would always be the best choice, and there would be no need to consider DBF. However, in the context of practical data assimilation problems, it is necessary to handle (critical to apply methods to) high-dimensional problems, such as weather forecasting, which often involve dimensions of ~10^6 or more and at the same time it is required to run in real-time. In this context, the Particle Filter is rarely used and is often not considered even as a comparison.
>
>     In the supplementary material, "Lorenz_latent_RMSE.png", "Lorenz_latent_testtime.png", "Lorenz_latent_traintime.png", "Lorenz_testtime_RMSE.png", and "Lorenz_traintime_RMSE.png" show the trade-off for DBF, whereas "RMSE_particles.png" show that for the PF. The comparison between the DBF and the PF is shown in "RMSE_inf.png". It is clear from "RMSE_inf.png" that DBF shows superior accuracy-computation trade-off compared to the PF.
>
>     The dimensionality of h can be either larger or smaller than that of z, but there is a trade-off: up to a certain latent dimensionality, increasing the dimension improves performance at the cost of longer computation time. Beyond that point, increasing the latent dimensionality no longer improves performance but only increases training time (although inference time remains relatively short compared to model-based approaches). Therefore, the optimal balance depends on the specific problem. For the Lorenz96 system, a dimensionality of 800 was a reasonable trade-off among 20, 80, 200, 800, and 2000 dimensions. As shown in the figure, the RMSE changes by only 7% (1.31 vs 1.23) in the range from 200 to 2,000 dimensions, indicating that the impact is not critical in this range.
>
>     The PF also has the trade-off. Although RMSE improves slowly as we increase the number of particles, the RMSE was poor (3.64) even with massively large number of particles (100,000) with very long inference time (2,000 seconds per initial condition)
>
> *While the paper comes with code for the DBF implementation...*
> - Thank you for pointing this out. We will add the derivation of the ELBO training objective in the appendix.
>
> *While ‘Strategy 1’ is introduced...*
> - We consider that the Strategy 1 is basically a linear dynamics problem with the training for data-driven Koopman operator. The moving MNIST problem demonstrates how the linear dynamics case works, therefore we have omitted this empirical evaluation.
>
> *Is your approach identifiable?...*
> - Yes, within the framework of the Koopman operator, the relationship between physical variables and latent variables is assumed to be 1:1. When the dimensionality of h is higher than that of z, the entire h-space is not utilized; instead, the trajectory on z is embedded within h-space.
>
> *Can DBF represent multimodal posteriors?*
> - Any non-Gaussian distribution can be represented by applying a nonlinear transformation on a Gaussian. Further study is needed to clearly state that the distribution can be learned in the current framework.
>
> *line 273: “computations for covariance matrices...*
> - See the comment directed to all reviewers.
>
> *One of your key motivations seems...*
> - The RMSE_inference figure shows that the inference time is very quick (0.1-0.2 seconds) for DBF with the latent dimensions in the range of [20, 2,000]. In contrast, PF requires the system simulation of the number of particles, therefore it gets significantly slower (2,000 seconds with 100,000 particles) if large number of particles are used.
>
> *Also, how do the state dimensions...*
> - We interpret the “state” in your question as referring to the latent state dimension h of DBF, which affects computational complexity. Please refer to the RMSE vs. latent dimension plot. In DBF, even with a 20-dimensional latent space, it already outperforms many other methods. Performance continues to improve with higher dimensions up to 800, but beyond that, increasing the dimensionality to 2,000 does not result in further improvement.
>
>     For the Particle Filter (PF), performance improves as the number of particles increases from 20 to 100,000. However, even with 100,000 particles, the performance was not particularly good. Moreover, this is the result for the 40-dimensional Lorenz 96 system; as the dimensionality increases to 100 or 400 dimensions, the performance of PF deteriorates catastrophically. In contrast, DBF can handle latent spaces with thousands of dimensions without issue:  you can see that from the fact that 40-dimensional Lorenz96 problem can be well assimilated with only 20 dimensional latent space.

---

> > ### Comment · Reviewer_b7Ma · 2024-11-26
> >
> > Thank you for your answers and the accuracy vs runtime analysis, this substantially strengthens the paper and I am raising my score accordingly.
> >
> > Please also update the manuscript with your answers, e.g. the ELBO derivation is still missing. Also the main points from your general response to all reviewers should be substantially incorporated into the paper, not just leaving the accuracy-compute trade off in the appendix for readers to randomly stumble across. In general you will make the paper easier to follow by referring directly to specific (sub)sections within the appendix (currently the manuscript just makes nonspecific references to "appendix" throughout).
> >
> > (NB- I think "ablation study" is a misnomer in this case, as it commonly refers to removing components of a single approach one by one to show how much of the overall performance is due to which, and that none can be omitted without significant loss of performance).
> >
> > Clarifications of some questions:
> >
> > > Is your approach identifiable?
> >
> > By "identifiable" I meant do you have guarantees/proofs that your approach will recover the _true_ latent dynamics? Compare e.g. ICA literature and disentangled representation learning e.g. [Yao et al. (ICLR 2022)](https://arxiv.org/abs/2110.05428).
> >
> > > Also, how do the _state_ dimensions vs the _observation_ dimensions affect the performance of DBF vs particle filters?
> >
> > Apologies for the confusion; by "state dimension" I meant the actual (true) latent state of the system (e.g. 8 for the moving digits), as opposed to the dimensions of the observation (1568 in this case). Your examples all seem to have very low-dimensional state.

---

> > > ### Comment · Reviewer_b7Ma · 2024-11-26
> > > **Applications of linear (unsupervised) vs nonlinear (supervised) DBF**
> > >
> > > My understanding is that DBF's nonlinear approach (requiring supervised training) will be used e.g. in assimilating real-world observations into weather simulators. Could you please also give some concrete applications examples for DBF's linear approach (not requiring supervision)?

---

> > > > ### Author Response · Authors · 2024-11-26
> > > >
> > > > As an example of a linear dynamics problem, we propose object tracking and self-localization in autonomous mobile robots. In self-localization, the robot’s position at the next time step can be estimated linearly via its velocity, elapsed time, and the current position. However, observations, such as distance sensor readings and RGB camera images, can vary nonlinearly depending on environmental factors, such as the presence of walls or the changes in the background of the image.
> > > >
> > > > Another example is object tracking. When the movement of an object in 3D space is described as uniform linear motion, the dynamics remain linear. However, the projection of this motion onto 2D space introduces nonlinearity due to the nature of the projection. In scenarios with multiple objects, further nonlinearity arises from occlusion. One potential approach is to combine an object detector, such as the YOLO model, with DBF. YOLO provides bounding boxes and confidence scores, and DBF has the flexibility (via $G_\theta$) to decide whether a new observation (e.g., a bounding box) is reliable. The YOLO confidence score can be used as input to $G_\theta$ (without additional training) to facilitate object tracking. We tested this straightforward method and observed that it significantly improves tracking performance compared to the conventional Kalman Filter.
> > > >
> > > > Alternatively, the model can be trained from scratch, enabling it to learn the confidence of observational information through the Inverse Observation Operator (IOO). This flexibility demonstrates the applicability of DBF to complex nonlinear observation scenarios.

---

> > > > > ### Comment · Reviewer_b7Ma · 2024-11-26
> > > > >
> > > > > Do you have any results on this object tracking setup? That would make for a substantially stronger paper - the moving MNIST digits is quite a toy setup, but a more realistic / real-world example would be pretty cool. Especially because in the real world the actual dynamics might only be approximately linear.

---

> > > > > > ### Author Response · Authors · 2024-11-26
> > > > > >
> > > > > > Thank you for your valuable feedback. We agree that evaluating the proposed method on more realistic scenarios enhances the paper's contribution. To address this, we have included an additional experiment on a real-world object tracking setup in Appendix C, with a reference to it provided in line 306, where we introduce the experiments. Due to space constraints, these results are presented in the appendix rather than the main text.
> > > > > >
> > > > > > We believe this experiment demonstrates the applicability of our method to more challenging and practical scenarios. Thank you for highlighting this important perspective.

---

> > > > > > > ### Comment · Reviewer_b7Ma · 2024-11-26
> > > > > > >
> > > > > > > Thank you. I would strongly encourage you to include the object tracking results in the main text, and move the toy MNIST example out of the main text to make space.
> > > > > > >
> > > > > > > If you can still answer my outstanding questions that would be appreciated.

---

> > > > > > > > ### Author Response · Authors · 2024-11-26
> > > > > > > >
> > > > > > > > Thank you for your feedback and suggestion. Following your recommendation, we have included the object tracking results in the main text and moved the MNIST example to the appendix to make space.

---

> > > > > > > > ### Author Response · Authors · 2024-11-27
> > > > > > > >
> > > > > > > > Let us answer the remaining questions.
> > > > > > > >
> > > > > > > > *Is your approach identifiable?*
> > > > > > > >
> > > > > > > > *By "identifiable" I meant do you have guarantees/proofs that your approach will recover the true latent dynamics? Compare e.g. ICA literature and disentangled representation learning e.g. Yao et al. (ICLR 2022).*
> > > > > > > >
> > > > > > > > DBF maximizes the ELBO rather than the marginal log-likelihood, $\log p(o_{1:T} , z_{1:T} )$. As long as the ELBO training objective aligns with the marginal likelihood, the parameters estimated through maximum likelihood estimation achieve asymptotic consistency with the true dynamics. However, if this alignment does not hold (i.e., there is a gap between the ELBO and the marginal log-likelihood), the argument for consistency is invalid, and the optimization instead results in parameters that maximize the approximate log-likelihood.
> > > > > > > >
> > > > > > > > DBF can represent the exact posterior distribution when the state-space model is linear Gaussian. In this scenario, the results coincide with those of the KF. As you can check by comparing equations (1), (2), and (4), (5), the KF posteriors can be reproduced by appropriately selecting the functions $f_{\theta}$ and $G_{\theta}$, such that $f_{\theta}$ becomes linear with respect to the observation and $G_{\theta}$ corresponds to a constant matrix. These correct functions can be obtained through ELBO optimization, provided that there are sufficient training samples.
> > > > > > > >
> > > > > > > > *Also, how do the state dimensions vs the observation dimensions affect the performance of DBF vs particle filters?*
> > > > > > > >
> > > > > > > > We are currently investigating the accuracy-compute trade-off for the double pendulum experiment and will update the PDF with the results by the update deadline.

---

> > > > > > > > ### Author Response · Authors · 2024-11-28
> > > > > > > > **A follow-up answer on identifiability**
> > > > > > > >
> > > > > > > > Let us follow-up on the question of identifiability:
> > > > > > > >
> > > > > > > > *Is your approach identifiable?*
> > > > > > > >
> > > > > > > > *By "identifiable" I meant do you have guarantees/proofs that your approach will recover the true latent dynamics? Compare e.g. ICA literature and disentangled representation learning e.g. Yao et al. (ICLR 2022).*
> > > > > > > >
> > > > > > > > If the question you are asking is whether the latent variables $h_t$ are identifiable in the sense that the probabilistic model assumed by DBF contains redundancy in the parameters, the answer is that they are not identifiable, as the latent space expressions in DBF may be redundant.
> > > > > > > >
> > > > > > > > Without this identifiability, it becomes challenging to interpret the values of the latent variables and the components of the dynamics matrix. However, we consider this lack of identifiability is not problematic in our problem setting. Our goal is to perform data assimilation (posterior estimation) given observational data and the (physical) SSM, rather than attempting to disentangle the latent space in which $h_t$ resides.

---

> ### Author Response · Authors · 2024-11-22
>
> *You mention Frerix et al....*
> - In data assimilation, observations are corrupted with noise, and not all physical variables can be directly observed. We think that probabilistic IOO is better than deterministic IOO as it naturally allows stochastic mapping between observation and physical variables.
>
> *How did you set up & run the baselines...*
> - We have used the open-source code (https://github.com/XiaoyuBIE1994/DVAE) for DVAEs. For KalmanNet, we have used (https://github.com/KalmanNet/KalmanNet_TSP).
>
> *Baseline methods could not handle...*
> - We show the results: in Lorenz96 with nonlinear observation experiment,
>     - With 115200 data (cf. KalmanNet consumes 120000): RMSE=2.11 (KalmanNet: 2.97)
>     - With 512000 data (cf. SRNN and VRNN consumes 512000): RMSE=1.49 (SRNN: 3.30, VRNN: 3.69)
>
>     However, we would like to politely question the meaning of comparing results against methods that diverge by stopping the training at the point where divergence occurs in the methods, especially since our methods incorporate architectural improvements to prevent divergence. In SRNN and VRNN, the instability of the training often results in outputs that are almost zero in all dimensions, even if the training did not crash.
>
> *line 353:...*
> - Taking the RMSE over four dimensions (two shapes and two spatial dimensions) for position and velocity over the last ten steps,
>     - (DBF) the positional RMSE: 0.39 $\pm$ 0.027, the velocity RMSE: 0.45 $\pm$ 0.042
>     - (EnKF) the positional RMSE: 6.3 $\pm$ 1.1, the velocity RMSE: 4.4 $\pm$ 2.2
>     - (ETKF) the positional RMSE: 5.7 $\pm$ 1.4, the velocity RMSE: 6.5 $\pm$ 8.4
>     - (PF) the positional RMSE: 4.8 $\pm$ 0.7, the velocity RMSE: 1.4 $\pm$ 0.23
>
> *line 365:...*
> - The most popular way to learn system parameters is to add the parameters in the states and do the estimate altogether. We have written the results in the table in Figure 2, which shows very poor performance.
>
> *I would have liked to see...*
> - We will add these in the Appendix.
>
> *line 480–485:...*
> - We utilize ensembles from the EnKF and ETKF methods to represent the posterior distributions. To assess the Gaussianity of these distributions, we apply the Shapiro-Wilk test to the ensemble members across all physical dimensions. The experiment considers 10 initial conditions, 80 time steps, and 40 physical dimensions, resulting in a total of 32,000 tests conducted for each level of observation noise. For EnKF, the number of distributions rejected as Gaussian at the 5% significance level is 1,684 for σ=1σ=1, 2,707 for σ=3σ=3, and 4,894 for σ=5σ=5. These results indicate that the degree of non-Gaussianity increases with higher observation noise.
>
>     For ETKF, the majority of ensemble distributions were rejected as Gaussian: 31,610 for σ=1σ=1, 31,616 for σ=3σ=3, and 31,616 for σ=5σ=5. This aligns with previous findings (e.g., Fig. 10 of Lawson & Hansen, 2004), which demonstrate that ETKF ensemble members tend to exhibit non-Gaussianity, particularly in nonlinear state space models.
>
>     reference: Lawson&Hansen (2004), *“Implications of Stochastic and Deterministic Filters as Ensemble-Based Data Assimilation Methods in Varying Regimes of Error Growth”*
>
> *I would have liked to see an ablation study*
> - See "Lorenz_latent_traintime.png" and "Lorenz_latent_RMSE.png". As we have explained above, 800 dimensions were good balance in terms of the accuracy and the training time.
>
> *line 90: ...*
> - We had 3DVar and 4DVar in mind. Indeed, the methodologies should be called point estimates. We will update the manuscript accordingly.
>
> *p.3, Eq. (1): ...*
> - Thank you very much for pointing this out. This should be $\mu_{t-1}$.
>
> *line 145: ...*
> - The virtual prior is the term $\rho(z_t)$. As you can see from the equation $r(z_t|o_t) = \frac{p(o_t|z_t)\rho(z_t)}{\int p(o_t|z_t)\rho(z_t)dz_t}$ in the text, $\rho(z_t)$ stands for a prior of $z_t$ for the inverse observation operator $r(z_t|o_t)$.
>
> *Eq. (6): ...*
> - For the first time step, the term is KL[q(h1|o1)|q(h1)]: we forgot to mention that q(h1|o0) is the initial distribution q(h1). Thank you for pointing this out and we will revise accordingly.
>
> *Would you call DBF part of the DVAE family...*
> - We regard DBF as a subclass of DVAE family. We will update the manuscript accordingly.
>
> *line 265:...*
> - We are not arguing that the LS4 approach is problematic. Here, we simply discuss the relationship between LS4 and our methodology.
>
> *in line 339 vs 346...*
> - The noise in line 339 is the observation noise in p(ot|zt) and the one in line 346 is the system noise in p(zt+1|zt). Our intention in line 346 is that we have tested with three different system noise levels and chosen the best value for EnKF, ETKF, and PF.

---

> ### Author Response · Authors · 2024-11-22
>
> *line 348: ...*
> - Indeed, memory consumption depends also on the sequence length and the batch size. We will revise the manuscript since the phrasing was not appropriate. Here, we have set the batch size to be one, and still the training was not possible. As was written in the text, the very high-dimensional observations make it impossible for KalmanNet to store all gradients into memory.
>
> *Fig. 2 (b):...*
> - Iterations is the training steps of NN (how many parameter updates the model has undergone).
>
> *Fig. 3 (b):...*
>
> *Table 1 and line 420...*
> - Yes, these are the averages of two $\theta$ and two $\omega$. We will revise the main text to clarify this point.
>
> *line 422,...*
> - Failed fractions: VRNN: 2/15 (13%), SRNN: 3/15 (20%), DKF: 2/15(13%). We will add these information in the text.
>
> *line 426: ...*
> - Yes it is.
>
> *lines 483, 485:...*
> - Thank you for pointing this out. You are correct, these two priors are both test prior distributions $q(z_t|o_{1:t-1})$. We will revise accordingly.
>
> We sincerely thank the reviewer for highlighting clarifications that improve the readability of our work.

---

### Official Review · Reviewer_6Lnv · 2024-11-04

**Soundness:** 3
**Presentation:** 3
**Contribution:** 2
**Rating:** 6
**Confidence:** 3

**Summary:**

This paper looks at state inference in non-linear state space models, that is, the problem of estimating latent "physical" variables z_t given observations o_t. The posterior of z_t given o_t is often non-Gaussian in models with a practical use, e.g., models of weather dynamics. This poses a challenge, and solutions have taken a path along using Gaussian approximations to the posterior or computationally intensive methods such as Sequential Monte Carlo. A key idea that the paper relies on is the introduction of additional latent variables (state expansion) h_t variables for which Gaussian inference is possible; these variables are later mapped to the original state variables and marginalized. In this sense, the method acts like a dynamic VAE but without the use of Monte Carlo approximations. Essentially, the assumption that the original physical variable dynamics can be linear in a sufficiently high-dimensional space when the mapping to that space is chosen correctly is taken. Two proposed training strategies are given, on the basis of a standard ELBO objective. Some standard examples with a Lorenz model and MNIST are presented to illustrate the method and good performance is shown.

**Strengths:**

Designing generic-purpose VAE for modeling time series is and will remain an important problem. The authors use nice illustrations to make their point. The use of the filter as opposed to the smoother is noted and that fact that it seems to be (more than) sufficient.

**Weaknesses:**

No idea of the computation time for the proposed method is given. Presumably, it should be faster? What if pretraining is taken into account? It is also unclear to me how to choose the dimensionality of the new state space h: is it of a comparable size to z_t or must it be much bigger? As the authors say, they need to know values of at least z_t.

**Questions:**

Can the method be used with approximate, even crude samples of z_t obtained by another method? Would it still add value there?

---

> ### Author Response · Authors · 2024-11-22
> **Thank you for the review: see Appendix E for the computational time**
>
> Thank you for your comments and the opportunity to address your concerns. We give answers to questions and comments you have raised in “weaknesses” and “questions” section.
>
> *No idea of the computation time for the proposed method is given*
> - The inference time is the central problem for practical applications. The (pre-)training time is therefore not a problem. A trained DBF gives inference for a batch in $\sim 0.13$ seconds.
> - On the other hand, model-based filtering approaches take longer and longer time as we increase the number of particles. For example, particle filter take 2,000 seconds to run with 100,000 particles.
>
> *It is also unclear to me how to choose the dimensionality of the new state space h*
> - **See Appendix E in the updated PDF file.** The dimensionality of h can be either larger or smaller than that of z, but there is a trade-off: up to a certain latent dimensionality, increasing the dimension improves performance at the cost of longer computation time. Beyond that point, increasing the latent dimensionality no longer improves performance but only increases training time (although inference time remains relatively short compared to model-based approaches). Therefore, the optimal balance depends on the specific problem. For the Lorenz96 system, a dimensionality of 800 was a reasonable trade-off among 20, 80, 200, 800, and 2000 dimensions. As shown in the figure, the RMSE changes by only 7% (1.31 vs 1.23) in the range from 200 to 2000 dimensions, indicating that the impact is not critical in this range.
>
> *As the authors say, they need to know values of at least z_t*
> - As we have written in the official comment to all reviewers, we focus on the problem of data assimilation (estimation of the posterior distribution) for a known state-space model. If the state-space model is given, it is possible to generate time series for o and z. Availability of the physical variable z during inference is not required. The problem we are solving involves finding an accurate posterior distribution under the given generative probabilistic model (state-space model). Such setting is typical in natural sciences including weather forecasting and ocean data assimilation. We believe that the applicability of our methodology is pretty wide.
>
> *Can the method be used with approximate, even crude samples of z_t obtained by another method? Would it still add value there?*
> - That differs from the problem setting we assume for data assimilation: in data assimilation, we do not require the actual samples of zt themselves, but rather the dynamics model that zt follows and the observation model that relates zt to the observations o. If this foundational state-space model is incorrect, the resulting state estimates will also be erroneous.

---

> > ### Comment · Reviewer_6Lnv · 2024-11-26
> >
> > Thank you for this: this clarifies the questions I had! I have updated my score.

---

### Official Review · Reviewer_mzw7 · 2024-11-04

**Soundness:** 3
**Presentation:** 3
**Contribution:** 2
**Rating:** 6
**Confidence:** 4

**Summary:**

This paper deals with data assimilation in general state-space models (SSM) and proposes a new method for approximating the flow of marginal state posteriors, also known as filter distributions. In the case of a linear Gaussian latent state process (but a general emission density), by formulating the filter recursion in terms of a so-called inverse observation operator (IOO)—which models the law of the current state given the corresponding observation—and approximating it by a Gaussian distribution parameterized by a neural network, Gaussianity is preserved through the filter recursion, allowing the generation of a sequence of approximate Gaussian filter distributions. The IOO is trained along with the unknown model parameters using variational inference, and in the case of linear dynamics this can be done without supervision. When the state process is nonlinear, the authors introduce another level of auxiliary latent states with linear Gaussian dynamics, generating the original latent state process through an auxiliary emission density. In this case, training cannot be performed without supervision, but requires access to training data consisting of a known history of both latent states and observations. The authors illustrate their method numerically with three examples, one where the goal is to identify images from the MINST dataset that move with linear dynamics and two where the goal is to estimate parameters and states in models with non-linear dynamics describing a double pendulum and a Lorenz system.

**Strengths:**

The idea of learning a Gaussian, NN-parameterized IOO to obtain Gaussian approximations of the filter distributions in the case of linear state dynamics is, to my knowledge, novel. Although I am more doubtful about the relevance of the method for models with nonlinear dynamics, at least in its current form, the simplicity of the proposed method appeals to me. Furthermore, the method's performance, at least in the linear example, is promising, although it is unclear how it generalizes to other models (as the assumed Gaussianity of the IOO introduces a bias that is not easily controlled; see my comment below). The article is fairly clearly written and the methodology is not too difficult to understand. So sum up, the proposed method should have some relevance, at least in the case where the latent-state dynamics is linear.

**Weaknesses:**

As far as I am concerned, the proposed methodology has three major shortcomings, which makes the paper boardeline.

(i) It seems crucial to use a Gaussian IOO, which is likely to lead to poor approximations for some models, e.g. in cases where the distribution of the current state is multimodal given the corresponding observation. Moreover, the bias implied by the assumed Gaussianity seems very difficult to control.

(ii) In the case of nonlinear dynamics, the method requires that training is done in a supervised manner, i.e. with access to the observed history of the latent state process, which is only possible in a limited number of contexts. However, in many cases, these states have no direct physical counterparts, but are only a phenomenological tool to capture patterns and features in the observed data (think for example of the unobserved volatility driving the dynamics of asset prices in a stochastic volatility model), preventing the method from being used. This significantly limits the applicability of the methodology.

(iii) To deal with the nonlinear case, the authors introduce an auxiliary state process (the h's), and a Gaussian emission density that generates the original states (the z's) given the auxiliary ones. My concern is that this construction strips the original state process of the Markov property (in the same way as the observations of an SSM are not Markov, even if the underlying states are so), leading to a model with a significantly different dynamics than the original one. Together with point (ii), this leads me to doubt even more the relevance of the method for the nonlinear case.

**Questions:**

(1) In the non-linear case, introducing auxiliary states, which fundamentally change the model and the methodology (e.g. by requiring supervised learning), seems to me a solution that is too ad hoc. Are there other ways in which nonlinearities in state dynamics could be addressed?

 (2) A similar question concerns the Gaussianity of the IOO. Is there any way in which a slightly more flexible distribution class could be used?

(2) If I have understood correctly, the numerical part evaluates the different filtering algorithms based on the RMSE with respect to the _true_ states? Since the goal of these algorithms is optimal filtering, i.e. to determine the posterior distributions of the states given the data and not the true, exact states, such a comparison makes it difficult, in my opinion, to claim that one algorithm is better than another. Would it not be better to compare the methods in terms of RMSE with respect to reference values calculated on the basis of a consistent algorithm, e.g. the particle filter with a massively large number (say, half a million) of particles.

**Details Of Ethics Concerns:**

No ethics concerns.

---

> ### Author Response · Authors · 2024-11-22
> **Thank you for the review!**
>
> Thank you very much for appreciating the novelty and simplicity of our method. We would like to argue that our methodology is very useful also in nonlinear dynamics case, ensuring the wide applicability of DBF. We write answers to the points you have raised in “weaknesses” and “questions”.
>
> *(i) It seems crucial to use a Gaussian IOO...*
>
> - In setting where the distribution of the current state is multimodal, we would use the auxiliary latent space for the assimilation even when the dynamics is linear. With that strategy, even if the posterior distribution over h is Gaussian, the distribution over z can be non-Gaussian because of the nonlinear transform from h. We think that this is sufficiently flexible.
>
> *(ii) In the case of nonlinear dynamics,...*
>
> - Here, we focus on the problem of data assimilation (estimation of the posterior distribution) for a known state-space model. If the state-space model is given, it is possible to generate time series for o and z. Availability of the physical variable z during inference is not required. The problem we are solving involves finding an accurate posterior distribution under the given generative probabilistic model (state-space model). We believe that such problems are significant in many real-world scenarios where the goal is to estimate the underlying state from observations based on physical laws.
>
> *(iii) To deal with the nonlinear case...*
> - It is true that the Markov property in z’s is lost but this does not mean the learned dynamics are significantly different. We have two reasons.
>
>     **The first one** is that our probabilistic model respects the Markov property in z’s in a certain limit. Assuming that
>     1. the noise of the emission model R in $p(z_t|h_t) = \mathcal{N}(z_t|\phi(h_t), R)$ goes to zero, $p(z_t|h_t) = \delta(z_{k} - \phi( z_{k}))$, and
>     2. Given 1. and if the model successfully learns the Koopman operator, $p(h_t|z_t) = \delta(h_{k} - \psi( z_{k}))$ (at least for h with the correspondent z),
>
>     With these assumptions,
>
>     $p(z_{t+1}|z_{1:t})$
>
>       $= \int p(z_{t+1}, h_{1:t+1} | z_{1:t}) dh_{1:t+1}$
>
>      $= \int p(z_{t+1}, | h_{1:t+1}, z_{1:t}) p(h_{1:t+1} | z_{1:t}) dh_{1:t+1}$
>
>      $= \int p(z_{t+1}, | h_{t+1}) p(h_{t+1} | h_{1:t}, z_{1:t}) p(h_{1:t} | z_{1:t}) dh_{1:t+1}$
>
>      $= \int p(z_{t+1}, | h_{t+1}) p(h_{t+1} | h_t) π_k δ(h_{k}-ψ(z_{k})) dh_{1:t+1}$
>
>      $= \int p(z_{t+1}, | h_{t+1}) p(h_{t+1} | ψ(z_t)) dh_{t+1}$
>
>      $= \int p(z_{t+1}, h_{t+1} | ψ(z_t)) dh_{t+1}$
>
>      $= p(z_{t+1}|z_t)$
>
>     The posterior distribution over $z_{t+1}$ depends only on the last timestep, $z_t$: therefore, z’s have the Markov property. In reality, the learned Koopman operator is not exact: therefore, finite noise of R is favored to reduce the evidence lower bound.
>
>
>
>     **The second reason** is that the learning process (i.e., the loss function) requires the model to reproduce the distribution of training data no matter what assumptions we use for the probabilistic model. For example, it is possible to directly predict the movement of rain clouds and estimate rainfall using, say, the most recent three time steps of satellite images. In this case, a third-order Markov model is used to predict the movement of rain clouds, which is different from the true dynamics of rain clouds. The difference in the Markov assumptions does not mean that the model cannot predict the movement of rain clouds.
>
> *(1) In the non-linear case,...*
> - We do not agree that the introduction of the auxiliary states fundamentally change the model and is an ad hoc solution. The key concept of DBF is to construct a computable expression for posteriors using Gaussian family and optimize the model via evidence lower bound. The latent variables h created by the model meets the criterion and it is able to express any nonlinear dynamics given sufficient latent dimensions.
>
>     The Gaussian assumption for the posteriors is commonly assumed in the state-of-the-art methods for large-scale data assimilation, e.g., LETKF and 4D-Var. Our methodology enriches the class of Gaussian posterior family by introducing the learnable parameters.
>
> *(2) A similar question concerns...*
> - Even if the posterior distribution over h is Gaussian, the distribution over z can be non-Gaussian because of the nonlinear transform from h. We think that this is sufficiently flexible.
>
> *(2) If I have understood correctly,...*
> - While we agree that comparison with the optimal filering is better, we have two reasons why the comparison against the true states is sufficient. Firstly, we take multiple initial conditions and average the results: the results for the optimal filtering would yield the best results in terms of RMSE with the true state.
>
>     Secondly, the benchmark problems (e.g., Lorenz96) are of high dimension and the particle filter of massively large number cannot give the optimal filtering result. Experiments with 100,000 particles took 2,000 seconds per initial condition.

---

> > ### Comment · Reviewer_mzw7 · 2024-11-26
> >
> > I thank the authors for addressing my comments. As I explained in my first review of the paper, I consider the paper to be the borderline, but the authors' clarifications have solidified my position to place it just above the acceptance threshold. However, I am not ready to raise this rating further.

---

### Author Response · Authors · 2024-11-22
**Thank you very much for the reviews: answers to core questions**

We sincerely thank the reviewers for their constructive feedback and are pleased that all reviewers unanimously agree on the significance of the problem our work addresses. We have carefully considered the comments and provide clarifications and additional insights regarding specific points raised.
## 1. The Need for True State Variables (z) During Training
We acknowledge the concern about the requirement for true state variables (z) in training and emphasize that **this is not a strong constraint in the scenarios we target**. Data assimilation is the task to estimate the filtering distribution (posterior) for a given state-space model (generative model). While generating the dataset assumes perfect knowledge of the state-space model and restricts the method's ability to learn its parameters, this problem setting is sufficiently prevalent in both natural sciences and industry. For example:
- Natural Sciences: Numerical simulation codes and observation models (e.g., sensor properties) are often available, enabling the generation of paired training data (z,o).
- Industry: In visual object tracking, it is possible to generate the synthesized 2D observation image sequences by simulating the moving objects in 3D space.

This methodology is exemplified in numerical weather forecasting, where the "Observing Simulation System Experiment" (OSSE) framework is a standard approach for evaluating data assimilation algorithms. Therefore, the requirement for (z,o) pairs during training aligns naturally with these practical use cases.

## 2. Training vs. Inference Time
We also wish to address concerns about training time. In real-world applications of data assimilation, inference time is the critical factor that dictates algorithm usability, especially in real-time settings. Computationally intensive methods like Particle Filters (PF) often become impractical due to the need for unrealistically large number of particles. In contrast:
Training time for DBF, while finite, is not a bottleneck as it occurs offline.
Once trained, DBF achieves fast inference (0.1~0.2 seconds in our Lorenz96 experiments) without requiring physical variables, enabling efficient real-time data assimilation only with observed data. On the other hand, The PF with 100,000 particles take ~2,000 seconds to complete the experiment for one initial condition.

The computational time (both for inference and training) scales linearly with the latent dimensions. The reason is discussed in the section below.

## 3. Trade-off Between Latent Dimensions and Performance
We address the inquiries from reviewers b7Ma and 6Lnv regarding the trade-off between latent dimensions and performance. Below, we provide empirical results to elucidate this relationship. The results are presented in Section E in the appendix.
### Experimental Results
We conducted experiments on the Lorenz96 model with nonlinear observation and observation noise of 1. The following configurations were evaluated:
- DBF: Latent dimensions of 20, 80, 200, 800, and 2,000.
- PF: Particle counts of 20, 200, 2,000, 20,000, and 100,000.

The results, **summarized in Section E in the appendix**, show that DBF achieves inference times of 0.1–0.2 seconds across all tested latent dimensions. Notably, DBF performs well even with a latent dimension of 20, demonstrating its effectiveness even when the latent dimension is reduced below the original state space dimensions.
DBF’s scalability and efficiency stand in stark contrast to PF, which incurs significantly higher inference times as the number of particles increases.
### Insights on Scalability
The efficient scaling of DBF is attributed to its parameterization of the dynamics matrix in the latent space. Specifically:

**Block-Diagonal Representation**:
  DBF models dynamics in the latent space (h)—obtained through a nonlinear transformation from the physical space (z)—using a block-diagonal dynamics matrix. This is feasible for matrices that are diagonalizable up to complex numbers. Given a sufficiently expressive nonlinear transformation, this representation guarantees the necessary representational power while maintaining computational efficiency.

**Linear Scaling with Latent Dimensions**:
  Increasing the dimensionality of the latent space corresponds to adding blocks to the dynamics matrix in parallel. This ensures that the computational cost scales linearly with the latent dimensionality. In contrast, a full matrix would require computational costs proportional to the cube of the dimensionality due to matrix inversion.

**Correlations in Physical Space**:
  While the latent variables are independent due to the block-diagonal structure, the transformation back to the physical space (z) allows for mutual correlations among physical variables. This ensures that the latent representation maintains sufficient flexibility to capture the interdependencies in the original system.

Below, we provide detailed answer to each comment.

---

> ### Comment · Reviewer_b7Ma · 2024-11-22
> **Manuscript update**
>
> You seem to have updated the submission, but the paper PDF appears to be the same as during initial review, and the supplementary material ZIP file now contains your additional results but no longer includes the code. I would kindly ask you to also upload your revised manuscript -- the additional figures should be part of the PDF, in main text or appendix as appropriate, whereas the supplementary material should include the code for reproducing the empirical results.

---

> > ### Author Response · Authors · 2024-11-25
> >
> > Thank you for your prompt response. We have replaced the PDF with an updated version that includes supplementary figures on the accuracy-computation trade-off in Appendix C, along with other updates to the main text as necessary (highlighted in blue). We are also continuing to work on further improvements to the manuscript.
> >
> > Additionally, we have moved the supplementary material back to the code, as the new figures are now included in the appendix.

---

> ### Author Response · Authors · 2024-11-26
> **Further manuscript update**
>
> We have made further updates to the PDF and continue to improve the paper as time allows.
> * As **b7Ma** pointed out, it is beneficial to reference the appendix from the main text where appropriate.
>     * We have included the derivation of the ELBO in the appendix and referred to this section from Section 2.5: “Training.”
>     * All instances of “see the appendix” have been updated to specify the relevant sections and subsections.
> * As **25ic** noted, our original text was misleading, potentially giving readers the impression that DBF requires the physical variables zt  for inference, thereby hindering real-time application. We have clarified this point on lines 208–209.
>     * Additionally, we have included a discussion on nested Monte Carlo (MC) sampling in Section A of the appendix, with a reference added on line 250, where we discuss the relationship between DBF and other DVAEs.
>
> We would like to extend our gratitude to the reviewers who have taken the time to carefully read through our updates and comments. Your thoughtful feedback has been invaluable in improving the clarity and quality of the manuscript.
>
> Updates from the original text, includeing the bullet points above, are highlighted with blue.
>
> We hope these updates enhance the manuscript's clarity and improve communication with the readers.

---

> ### Author Response · Authors · 2024-11-27
> **Manuscript update**
>
> We have included the accuracy-computation trade-off figure for the double pendulum in Section E.1, along with the sample trajectory for DBF and PF in Section C.2 of the appendix. Due to the lower physical state dimensions, the difference in the accuracy-computation trade-off between DBF and PF was smaller than in the Lorenz96 experiment. With a sufficiently large number of particles (20,000 and 100,000), the PF successfully estimates the state variables $\theta$. However, estimating the angular velocities $\omega$ is more challenging than $\theta$, as $\omega$ is not directly observable and requires the models to integrate information across time steps. The performance of the PF nearly saturates at 20,000 and 100,000 particles, while DBF achieves comparable results when $dim(h_t)$ exceeds 20.
>
> The accuracy-computation trade-off figure for the PF in the Lorenz96 problem (Figure 21) has been slightly updated due to an incorrect noise level in the previous version. After correction, we observe a moderate improvement in the PF's performance. Nevertheless, the accuracy-to-computation trade-off for the DBF remains significantly superior to that of the PF.

---

> ### Author Response · Authors · 2024-11-28
>
> We have made a few minor updates on the PDF file:
>
> - The bullet point on linear dynamics in the introduction has been revised to reflect the inclusion of the object tracking example in the main text.
> - Discussions on additional references (NKF, KVAE, EKFNet, Auto-EnKF) have been added.
> - The object tracking section has been modified for improved clarity.

---

### Meta-Review · Area_Chair_jtZ5 · 2024-12-19

**Metareview:**

The paper 'Deep Bayesian Filter for Bayes-Faithful Data Assimilation' was reviewed by 4 reviewers who gave it an average score of 5.5 (final scores 5+5+6+6). The reviewers found the paper relevant, appreciated the experiments, and easy to follow. On the other hand, the reviewers had concerns related to the practical aspects of the approach, insufficient evaluations, and lack of details. Some of these issues were addressed during the discussion phase. However, as the paper now stands it remains borderline, and the right decision appears to be to reject the paper in its current form to allow the authors to improve their work.

**Additional Comments On Reviewer Discussion:**

This paper was subject to active discussion during the author-reviewer discussion phase. The average score increased from 4.75 -> 5.5 during the discussion.

---

### Decision · Program_Chairs · 2025-01-22

Reject